# The impact of climate change on economic output across industries in Chile

**Karla Hernandez[1], Carlos Madeira** [2]*

**1** University of Wisconsin-Madison, Madison, WI, United States of America, **2** Central Bank of Chile, Santiago, Chile

* cmadeira@bcentral.cl

## Abstract

Using region-industry panel data for Chile over the period 1985 to 2017, we find no effect of precipitation changes on GDP and a negative impact of higher summer temperatures on Agriculture-Silviculture and Fishing. An increase of one Celsius degree in the month of January implies a 3% and 12% GDP reduction in Agriculture and Fishing, respectively. There is also a negative effect of higher temperatures in January on Construction and Electricity, Gas, and Water. Our analysis suggests that climate change did not have a big impact on the Chilean economy during this period. Stress test exercises that select only the negative and statistically significant coefficients imply that the Chilean GDP would fall between -14.8% and -9% in 2050 and between -29.6% and -16.8% in 2100, according to our model.

**Data Availability Statement:** We published all the data in Mendeley Data: https://data.mendeley.com/datasets/zyrdg56hzr/1 doi: 10.17632/zyrdg56hzr.1.

**Funding:** The author(s) received no specific funding for this work.

## Introduction

Climate change is predicted to affect negatively the economic growth of almost all the countries across the world [1–3]. Since the negative consequences fall disproportionately on the poorest countries due to their proximity to the earth's Equator, the impact on the average world GDP per capita could be as high as -20% [4]. It is estimated that the Latin America region will suffer substantially from global warming in the 21st century, with some Caribbean countries being strongly affected due to their oceanic location and dependence on the agriculture and fishing sectors [5–7]. Due to its worst impact on the poorest countries [8] and the poorest households, climate change will be a significant threat to economic growth and reducing income inequality in Latin American countries [7, 9]. Empirical estimates show that global warming reduced the GDP per capita of the poorest countries by 17–31% over the last half century, making it more difficult for poorer nations to converge towards developed economies and increasing inequality between countries [8].

This study provides a view of the economic impact of climate change in Chile over the past 35 years, focusing on its impact across different industries and regions. In the literature it has been challenging to provide systematic evidence that rising temperatures affect the growth rate of economic activities beyond sectors that are naturally exposed to outdoor weather conditions such as agriculture, fishing and construction [10–13]. Our work presents a contribution relative to Colacito et al. (2019) [13], who make a similar analysis for the

**Competing interests:** The authors have declared that no competing interests exist.

USA across states and industries. Our work advances upon the previous literature by showing a similar analysis for Chile. Chile is an interesting case, because it is a developing economy with a much stronger relevance of the primary sectors in its output and it is located in the southern hemisphere which will be differently affected by climate change relative to the north [14, 15].

Using annual frequency GDP data for 12 economic sectors across 15 regions of Chile over the period 1985 to 2017, we find that temperature and rain precipitation fluctuations had little impact on economic activity, except for the Agriculture-Silviculture and Fishing sectors. Our econometric model has different coefficients for each industry and it includes as control variables the temperature and precipitation for each season (whether quarterly seasons or months) plus the industry-region growth lag, time fixed-effects at the year level, and fixed-effects for the regions. The model therefore accounts for both unobserved macroeconomic shocks affecting each industry and unobserved heterogeneity at the region-industry level.

Most studies for the impact of climate change on GDP use international level datasets with GDP for many countries and information on their temperatures and precipitation [2, 12, 16]. In this work, however, we use a dataset that is specific for Chile and its regions-industries. Therefore we apply a methodology similar to Colacito et al. (2019) [13] who also use state-industry data specific to the USA, finding that higher summer temperatures affected negatively the economic output of at least half of the industries, especially finance, insurance and real estate. Using annual GDP growth data across 12 industries for each of the 15 Chilean regions over the last 35 years, we find a statistically significant impact of climate change during the Summer season for the Agriculture and Fishing sectors. Each Celsius degree of temperature increase in the months of January implies a GDP reduction of 3% and 12% for the Agriculture and Fishing sectors, respectively. However, many industries have either been unaffected or could even be getting a positive impact from the temperature increases implied by climate change. Furthermore, some impacts of the temperature increase can be positive for the economic output outside of the Summer months.

Our main econometric results imply a low impact of climate change in Chile over the past 35 years. This is consistent with the previous literature showing that Chile so far has received little impact from climate change in terms of overall climate change costs [17], GDP costs [18], temperature fluctuations [2, 19], water availability [20] or labor hours lost to high temperatures [21]. Furthermore, studies such as Dell et al. (2012) [16] have found little effect so far of climate change in high income economies such as Chile, with negative effects being significant only for poorer nations.

This work is related to the research on the economic costs of climate change, with a particular emphasis on GDP [4, 12] and physical risks. Previous research for Chile shows that electricity generation and manufacturing are the sectors with the highest carbon emissions in Chile, but using an input-output framework then the manufacturing and mining sectors (in particular, their exports) are the highest indirect sources of carbon emissions [22]. Finally, climate change issues are gaining more relevance in Chile, as pension funds prefer investments with higher Environmental, Social and Governance (ESG) factors [23] and lower reliance on fossil fuels which are heavily used in Latin America [24]. Finally, Hernández and Madeira (2021) [25] show a literature review about the impact of climate change in Chile in a wide range of aspects, from GDP to water availability and migration. This article is organized as follows. Section 2 details the data used and the econometric methodology. Section 3 comments on the empirical findings over the 1985–2017 period, while Section 4 summarizes the conclusions and implications for policy.

## Methods & data

### Regional-industry GDP data

We use region-industry level GDP series for Chile over the period 1985–2017 from the National Accounts data publicly available from the Central Bank of Chile [26]. There are 12 industries shown in Table 1. Prior to 2007, Chile was divided into 13 regions. In 2007, region I split into regions I and XV and region X split into regions X and XIV, resulting in 15 regions. Region XIII is particularly important, being the Metropolitan Region of the capital Santiago, which represents around 40% of the national GDP and population.

For the years 1985–2007, we created a 15 region and 12 industry panel series assuming that each of the divided regions I and X shares of industry-level GDP is constant between 1985–2008. We allocate the industry-level GDP of regions I and X across their future sub-divided regions according to their share of the combined region's GDP for each industry in the year 2008:

$$GDP_{r,i,t} = GDP_{A(r),i,t} \frac{GDP_{r,i,2008}}{GDP_{A(r),i,2008}}, \text{ for } t \leq 2007, \tag{1}$$

with $t$ representing the year, $i$ the industry, $r$ being the region classification after 2008 and $A(r)$ being the region classification before 2008. In particular, $A(r) = I + XV$ for $r = I, XV$ and $A(r) = X + XIV$ for $r = X, XIV$. This adjustment is possible because the original region classification for the period 1985–2007 was exactly the same as the recent 2008–2017 classification for regions II, III, IV, V, VI, VII, VIII, IX, XI, XII and XIII. However, the original regions $I$ and $X$ in the period 1985–2007 are exactly equivalent to the sum of regions $I$ and $XV$ and the sum of regions $X$ and $XIV$, respectively, for the period 2008–2017. In S2 Appendix at the end of the article, we also show results with a set of 13 regions over the entire period 1985 to 2017.

The data is reported in four separate series 1985–1996, 1996–2003, 2003–2008, and 2008–2017 with base years 1986, 1996, 2003, and 2013 respectively. We harmonize the data as follows. To join adjacent series, we use the common year that is available in the series of both base years to create an adjustment factor for each region-industry observation as the ratio of GDP measured in the more recent base year to the GDP measured in the previous base year: $Adj_{r,i,b=t_0} = GDP_{r,i,t^*,b=t_1}/GDP_{r,i,t^*,b=t_o}$, with $r$ denoting the region, $i$ the industry, $t_1$ being the most recent base year and $t_0$ the previous base year. $t^*$ is the first period in the new base year $t_1$ series and the last period for the old base year $t_0$ series. We multiply each observation in the

Table 1. Fraction (%) of the value of each industry in national GDP (1985–2017).

| Industry Code | Industry Name | % of GDP |
|:---:|:---:|:---:|
| 1 | Agriculture and Forestry | 3.5 |
| 2 | Fishing | 0.5 |
| 3 | Mining | 15.1 |
| 4 | Manufacturing | 12.7 |
| 5 | Electricity, Gas, and Water (EGA) | 4.0 |
| 6 | Construction | 6.8 |
| 7 | Commerce, Restaurants, and Hotels | 10.9 |
| 8 | Transport and Communications | 8.2 |
| 9 | Financial Services | 14.9 |
| 10 | Home Ownership | 8.2 |
| 11 | Personal Services | 11.5 |
| 12 | Public Administration | 5.2 |

earlier dataset of base year $t_0$ by the adjustment factor: $Adj_{r,i,b=t_0}$. This procedure was applied first to link the most recent 2008–2017 series to the previous series 2003–2008, then to the series 1996–2003 and the series 1985–1996, to finally obtain the combined series 1985–2017. Therefore the adjusted GDP series used in the article are:

$$GDP_{r,i,t}^{adj} = GDP_{r,i,t}, \text{ for } t \in [2008, 2017], \tag{2a}$$

$$GDP_{r,i,t}^{adj} = GDP_{r,i,t}Adj_{r,i,b=2003}, \text{ for } t \in [2003, 2007], \tag{2b}$$

$$GDP_{r,i,t}^{adj} = GDP_{r,i,t}Adj_{r,i,b=2003}Adj_{r,i,b=1996}, \text{ for } t \in [1996, 2002], \tag{2c}$$

$$GDP_{r,i,t}^{adj} = GDP_{r,i,t}Adj_{r,i,b=2003}Adj_{r,i,b=1996}Adj_{r,i,b=1985}, \text{ for } t \in [1985, 1995]. \tag{2d}$$

We create datasets of both nominal and real GDP where the real GDP is adjusted using the UF or the value of real monetary unit which is indexed to the CPI [27]. Dividing the nominal GDP series by the UF value results in a real series for Chile. The average UF data for each year is also publicly available from the Central Bank of Chile, based on daily UF values published by the Chilean Bureau of Official Statistics (INE, *Instituto Nacional de Estadísticas* in Spanish). The UF money index is commonly used by companies and individuals in Chile for all kinds of long term contracts, including loans, real estate purchases, rent and wages. This option is made, because there are no price series in Chile that are valid for different industries in order to obtain real quantities per industry. Table 1 shows the average value of each industry in terms of the national GDP over the period 1985 to 2017. The largest economic sectors are Mining (15.1%), Financial Services (14.9%), Manufacturing (12.7%), Personal Services (11.5%) and Commerce (10.9%), with shares between 10.9% and 15.1% of the national GDP over the last 35 years.

In some years, industry GDP is negative for specific regions. This occurs in regions where few firms are operating in that sector. For example, negative values for industry-region GDP can occur in years in which a firm has costs exceeding revenue on its balance sheet. We replace negative values with 0 when calculating the growth rate of industry-region level GDP, therefore growth rates can be either -100% or missing when one of the years has a zero output value. Using weighted regressions for the value of each region-industry is also an adequate way to solve this, because those observations are attributed a zero weight. Table B1 in S2 Appendix shows that the results are robust to using weighted regressions. Furthermore, since 2008 the Central Bank of Chile computes the share of financial services costs for each region-industry. Before 2008 the financial services costs are reported for each region, but are not disaggregated to the industry level, implying that the output of each industry is slightly over-estimated before 2008.

Table 2 shows the average GDP of each industry for each of the 15 regions over the period between 1985 and 2017, which shows large disparities across regions. For example, Mining represents a share close to 0% of the regions VIII, IX, X and XIV, plus a share between 1.7% and 3.7% for the regions VIII, XI, XIII and XV. However, Mining represents more than 50% of the GDP in regions II and III, and also has a share between 15.7% and 37.1% of the value in regions I, IV, V and XII. Therefore Mining is the largest economic sector in Chile, but its resources are unequally distributed across regions. The capital region (XIII, Metropolitan Region of Santiago) represents more than 40% of the national GDP and population, being particularly important. For the capital region XIII the top industries are Financial Services (23.3%), Commerce (17.4%), Personal Services (12.7%) and Manufacturing (12.3%). Therefore

**Table 2. Fraction (%) of the value of each industry for the GDP of each region (1985–2017).**

| Industry / Region | I | II | III | IV | V | VI | VII | VIII | IX | X | XI | XII | XIII | XIV | XV |
|---|---|---|---|---|---|---|---|---|---|---|---|---|---|---|---|
| Agriculture and Forestry | 0.1 | 0.0 | 1.8 | 7.0 | 3.0 | 11.7 | 12.7 | 7.4 | 14.5 | 7.7 | 3.7 | 1.9 | 0.9 | 13.3 | 5.3 |
| Fishing | 1.4 | 0.2 | 0.6 | 0.4 | 0.2 | 0.0 | 0.1 | 1.2 | 0.3 | 5.9 | 16.6 | 1.0 | 0.0 | 1.0 | 2.4 |
| Mining | 37.1 | 63.2 | 50.5 | 30.1 | 15.7 | 30.7 | 1.7 | 0.1 | 0.0 | 0.0 | 3.7 | 19.5 | 2.2 | 0.0 | 2.7 |
| Manufacturing | 7.5 | 5.8 | 1.8 | 2.9 | 16.6 | 10.4 | 13.3 | 22.8 | 11.4 | 20.9 | 7.5 | 25.3 | 12.3 | 25.7 | 12.6 |
| EGA | 1.7 | 2.6 | 5.4 | 2.2 | 3.3 | 5.2 | 17.5 | 9.3 | 1.7 | 4.1 | 1.2 | 1.9 | 2.6 | 4.3 | 1.6 |
| Construction | 7.5 | 6.7 | 9.2 | 7.8 | 7.5 | 7.1 | 8.2 | 7.1 | 8.6 | 8.0 | 8.1 | 5.8 | 6.2 | 5.2 | 6.4 |
| Commerce | 8.9 | 2.7 | 3.9 | 7.4 | 6.7 | 6.4 | 6.3 | 6.3 | 8.4 | 7.9 | 7.4 | 6.0 | 17.4 | 8.0 | 8.0 |
| Transportation | 7.6 | 4.6 | 5.0 | 6.9 | 12.9 | 5.1 | 7.6 | 8.8 | 8.1 | 9.0 | 8.6 | 7.0 | 8.7 | 7.6 | 16.0 |
| Financial Services | 8.8 | 8.9 | 12.9 | 9.5 | 8.6 | 8.0 | 6.5 | 7.5 | 7.5 | 8.2 | 8.6 | 9.1 | 23.3 | 7.0 | 8.4 |
| Home Ownership | 4.1 | 1.9 | 4.2 | 7.9 | 10.3 | 6.5 | 10.1 | 9.3 | 10.6 | 7.7 | 6.1 | 6.2 | 9.8 | 8.2 | 8.9 |
| Personal Services | 7.0 | 3.7 | 5.8 | 11.5 | 11.8 | 11.0 | 14.3 | 14.0 | 19.7 | 15.2 | 12.8 | 8.1 | 12.7 | 15.6 | 17.2 |
| Public Administration | 4.8 | 1.4 | 4.0 | 5.9 | 5.9 | 3.6 | 6.7 | 6.3 | 9.7 | 8.5 | 17.4 | 9.9 | 4.7 | 8.8 | 16.7 |

it is interesting to observe that both Financial Services and Commerce are much more prevalent in the capital region than at the national level, while Mining has almost no value for the capital even if it represents the largest economic sector in the nation.

## Weather data

We use weather data available from the University of Delaware Air Temperature and Precipitation dataset [28] that provides gridded mean monthly surface air temperature (in Celsius degrees) and total monthly precipitation (in centimeters per month) data from 1900–2017. The data covers the terrestrial area of the globe with a grid size of 0.5 degree latitude x 0.5 degree longitude, which is approximately 56km x 56km at the equator. The grid squares intersect the area of Chile. We use geospatial software (QGIS) to aggregate the weather data to the regional level. First, we determine the fraction of each grid that falls within the borders of each region. Then, for each region, we create the regional weather series as a weighted average of the weather series of each grid where the weight is equal to the share of the grid that intersects the region. In this way, a grid square that has $\frac{1}{2}$ of its area intersecting a region receives $\frac{1}{2}$ the weight of a grid square that completely intersects the region. The joint panel dataset of weather variables and GDP by region-industry is publicly available in Madeira (2022) [27].

Fig 1 shows the yearly precipitation and temperature from the University of Delaware data for Chile between 1950 and 2017, reporting the minimum, maximum and mean monthly values over the 12 months of the year. The national-wide temperature and precipitation are reported as weighted averages of the regions by surface area or by the GDP of each region. Both measures differ, since some regions can be large in surface area (square kilometers), but small in terms of GDP and economic activity (or in terms of population). Calculating average precipitation weighted by each region's GDP can help obtain more accurate measures for the temperature and precipitation that affect economic activities such as agriculture, transports and services. Surface weights on the other hand may be a better proxy for the impact of weather changes on natural habitats and biodiversity. Minimum and mean precipitation is larger for the weighted surface measure, because larger regions have higher precipitation. Maximum and mean temperatures are lower by weighted surface, since the larger regions are cooler. Both the weighted GDP and weighted surface measures at the national level show that —despite large fluctuations in some years—the mean precipitation has been falling in Chile over time, while mean temperatures have increased substantially.

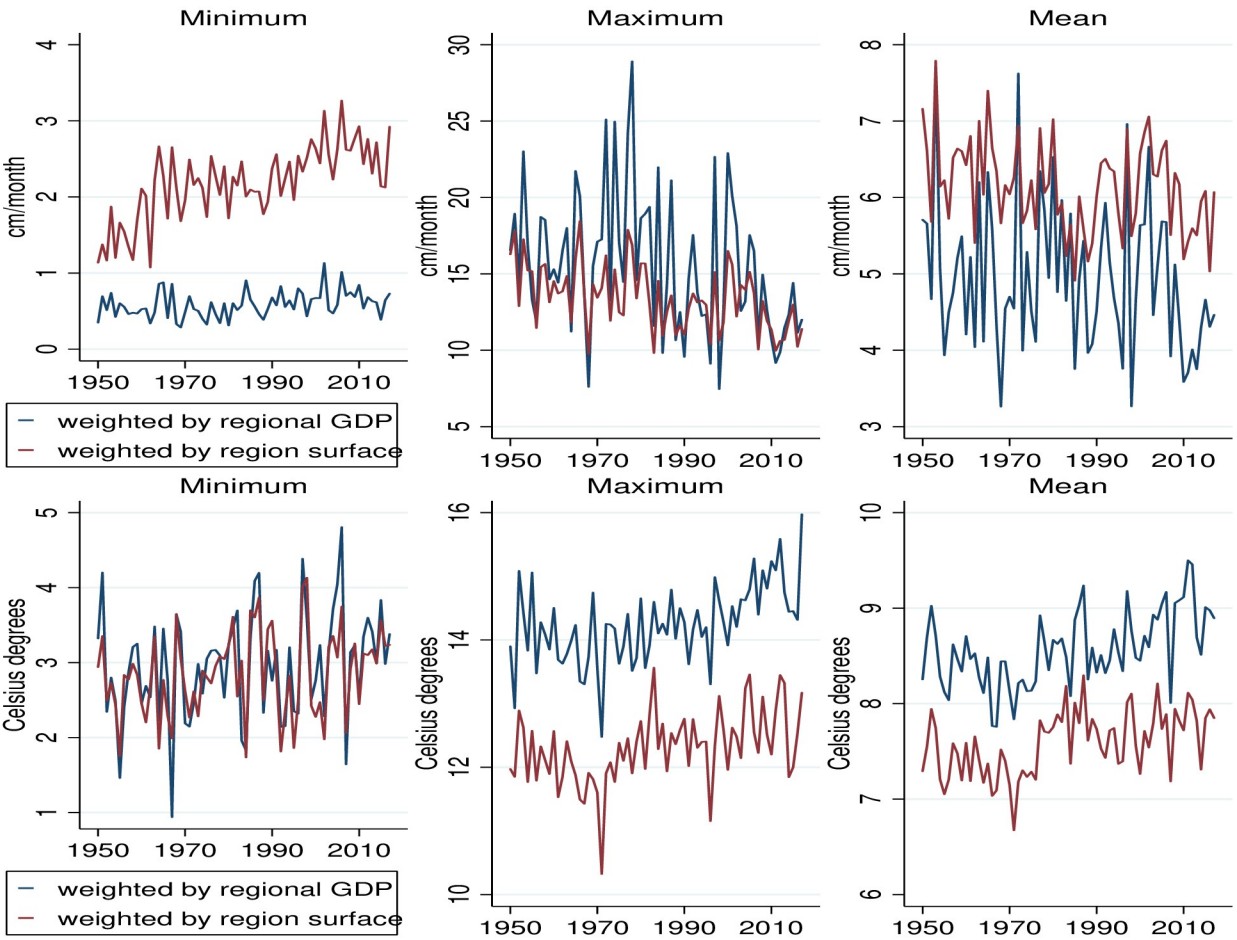

**Fig 1. The evolution of the yearly precipitation and temperature (weighted by the regional GDP in 2017 or by the surface area of each region) during the period 1950–2017.** Minimum, Maximum and Mean values are from January to December of each year.

To summarize the regional heterogeneity in a more succinct way, we create 4 macrozones, with macrozone 1 "North Chile" corresponding to regions I, II, III, IV and XV, macrozone 2 "Central Chile" corresponding to regions V, VI, VII, VIII, macrozone 3 "South Chile" corresponding to regions IX, X, XI, XII and XIV, and macrozone 4 "Metropolitan Region" corresponding to region XIII (which concentrates around 45% of the population and GDP of the nation). Fig 2 shows the yearly precipitation and temperature for each macrozone between 1950 and 2017. For simplicity we report only the weighted values by surface area. Fig 2 shows that mean precipitation has been falling in the Central, South and Metropolitan macrozones, while mean temperatures have been increasing across all the macrozones. Fig D1 in S4 Appendix shows a similar qualitative pattern in the temperature and precipitation values weighted by GDP for each macrozone.

Table 3 summarizes how much the distribution of the precipitation and temperature in Chile and its macrozones changed between 1950 until 1985 and between 1985 until 2017. Since there are substantial fluctuations between individual years, we implement a comparison by decades between 1950–1959, 1980–1989 and the last 7 years between 2010–2017. The results in Table 3 show that the mean precipitation weighted by surface are decreased substantially between 1950–1959 and 1980–1989 and also decreased slightly between 1980–1980 and

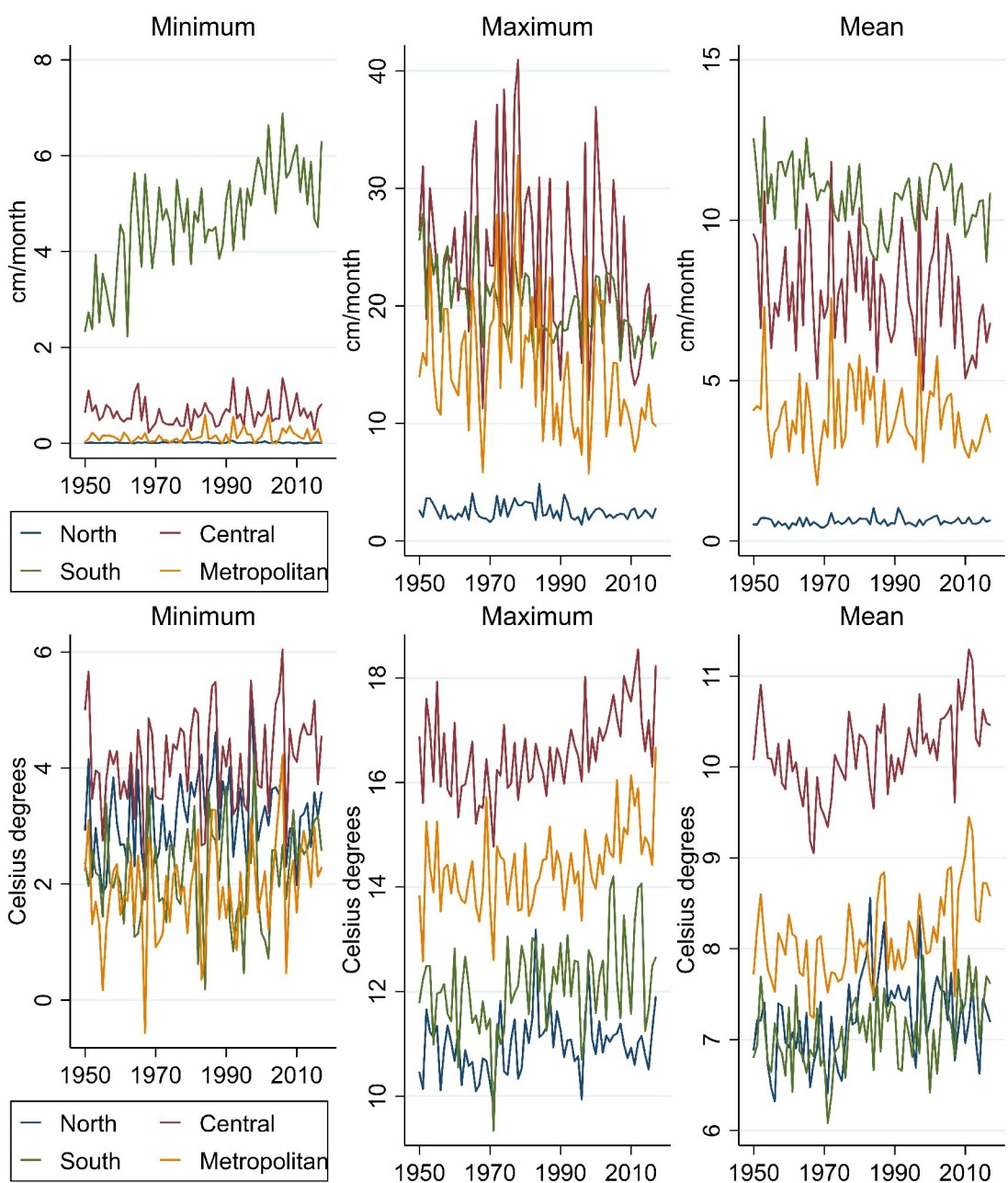

**Fig 2. The evolution of the yearly precipitation and temperature (weighted by the surface area of each region) for each macrozone during the period 1950–2017.** Minimum, Maximum and Mean values are from January to December of each year.

2010–2017. The results are similar for mean precipitation weighted by regional GDP, but with a sharper fall in mean precipitation between 2010–2017. Maximum precipitation in Chile also decreased substantially between 1950–1959 and 1980–1989 and again between 1980–1980 and 2010–2017, whether with surface or GDP region weights. The results also show a substantial decrease in maximum and mean precipitation across all macrozones (with either surface or GDP weight), except for the North, although minimum precipitation changed only slightly (except for the South).

**Table 3. Changes yearly minimum, maximum and mean in precipitation and temperature.**

| Period | Macrozone | Precipitation | | | Temperature | | |
|---|---|---|---|---|---|---|---|
| | | Min | Max | Mean | Min | Max | Mean |
| *Regional surface weights* | | | | | | | |
| 1989–1959 | Chile | 0.64 | -2.19 | -0.84 | 0.41 | 0.37 | 0.43 |
| 2017–1989 | Chile | 0.49 | -1.70 | -0.06 | -0.02 | 0.13 | -0.03 |
| 1989–1959 | North | 0.00 | 0.20 | 0.05 | 0.88 | 0.59 | 0.81 |
| 2017–1989 | North | -0.01 | -0.45 | -0.03 | -0.63 | -0.51 | -0.63 |
| 1989–1959 | Central | -0.14 | -0.67 | -0.56 | 0.20 | -0.21 | -0.03 |
| 2017–1989 | Central | 0.11 | -6.90 | -1.54 | 0.19 | 1.11 | 0.52 |
| 1989–1959 | South | 1.46 | -4.86 | -1.77 | 0.06 | 0.35 | 0.24 |
| 2017–1989 | South | 1.06 | -1.10 | 0.40 | 0.45 | 0.37 | 0.33 |
| 1989–1959 | Metro | 0.02 | -0.57 | 0.11 | 0.30 | 0.18 | 0.16 |
| 2017–1989 | Metro | 0.01 | -5.27 | -1.08 | 0.31 | 1.07 | 0.67 |
| *Regional GDP weights* | | | | | | | |
| 1989–1959 | Chile | 0.02 | -0.79 | -0.28 | 0.29 | 0.16 | 0.20 |
| 2017–1989 | Chile | 0.08 | -4.50 | -0.89 | 0.13 | 0.73 | 0.37 |
| 1989–1959 | North | 0.01 | 0.18 | 0.05 | 0.79 | 0.52 | 0.74 |
| 2017–1989 | North | -0.01 | -0.43 | -0.04 | -0.60 | -0.47 | -0.60 |
| 1989–1959 | Central | -0.12 | -0.21 | -0.38 | 0.12 | -0.13 | -0.01 |
| 2017–1989 | Central | 0.08 | -6.53 | -1.38 | 0.21 | 1.01 | 0.51 |
| 1989–1959 | South | 0.44 | -5.11 | -2.41 | -0.23 | 0.11 | -0.07 |
| 2017–1989 | South | 0.61 | -3.32 | -0.30 | 0.47 | 0.72 | 0.44 |
| 1989–1959 | Metro | 0.02 | -0.57 | 0.11 | 0.30 | 0.18 | 0.16 |
| 2017–1989 | Metro | 0.01 | -5.27 | -1.08 | 0.31 | 1.07 | 0.67 |

*Notes*: Precipitation in centimeters per month, and temperature in Celsius. Between the averages for 1950–1959 and 1980–1989 and between the averages for 1980–1989 and 2010–2017. Results for Chile and each macrozone weighted by region's surface area and regional GDP in 2017.

In terms of temperature changes, Table 3 shows a substantial increase in mean temperature between 1950–1959 and 1980–1989 with regional surface weights, although barely no change afterwards. However, with regional GDP weights there was an increase in mean temperature of 0.20 Celsius between 1950–1959 and 1980–1989 and an even stronger increase of 0.37 Celsius between 1980–1989 and 2010–2017. The maximum temperature for Chile also increased substantially between 1980–1989 and 2010–2017 with increases of 0.13 and 0.73 Celsius degrees with surface and GDP weights, respectively. Again, the North macrozone differs from the others in the sense that it experienced a large increase of 0.81 Celsius degrees between 1950–1959, followed by a strong decrease of -0.63 Celsius between 1980–1989 and 2010–2017. The Central, South and Metropolitan macrozones experienced a strong increase in mean and maximum temperatures in the recent period between 1980–1989 and 2010–2017. Mean temperatures for the Central, South and Metropolitan macrozones increased, respectively, by 0.52, 0.33 and 0.67 Celsius degrees between 1980–1989 and 2010–2017 with surface weights, while maximum temperatures in the same macrozones increased by 1.11, 0.37 and 1.07 Celsius degrees. When weighted by regional GDP, the results for the period between 1980–1989 and 2010–2017 are very similar. With GDP weights, the mean temperatures increased by 0.51, 0.44 and 0.67 Celsius degrees for the Central, South and Metropolitan macrozones, respectively, while the maximum temperatures increased by 1.01, 0.72 and 1.07 Celsius degrees. Overall, the results in Table 3 document a decrease in precipitation and increase in temperatures for Chile and all its macrozones (except the North) between 1980–1989 and 2010–2017.

## Econometric model

It is well known that temperature affects the dynamics of virtually all chemical, biological and ecological processes [12], while precipitation can affect agriculture [5, 29], especially in Latin America [7]. and also non-agricultural activities if excessive floods disrupt transport and urban connections [11, 12]. Chile, in particular, has been strongly affected in terms of reduced water availability [20] and a decade long mega-drought [25]. Zivin and Neidell (2014) [30] found that warmer temperatures reduce labor supply, while Cachon, Gallino, and Olivares (2012) [31] document that high temperatures decrease productivity and performance.

Seasonal temperatures and precipitation can affect productivity both in outdoor activities such as agriculture, fishing and construction [7, 11], but also for non-agricultural activities due to the influence of the weather on workers' health or urban movement [12, 13]. For this reason our vector $T_{r,s,t}$ for the measure of the weather variables in region $r$ in season $s$ of year $t$ includes both average temperature and precipitation.

There can be other shocks besides the weather (for instance, international shocks such as the Great Financial Crisis or higher demand from commodities due to a higher economic growth in China) that affect the economic growth of each industry $i$ at time $t$. For this reason our chosen model must account for both time-industry fixed-effects ($\alpha_{t,i}$) and the dynamic effect of shocks in the previous year by controlling for the lagged growth ($\Delta y_{r,i,t-1}$). Furthermore, an adequate model must account for regional heterogeneity in terms of natural resources, weather and industry specialization, therefore our model will include fixed-effects across regions and industries ($\alpha_{r,i}$) and heterogeneous coefficients ($\beta_{s,i}$ for the impact of the weather variables $T_{r,s,t}$, $\rho_i$ for the impact of the lagged growth $\Delta y_{r,i,t-1}$).

Our econometric model therefore follows a panel structure of log GDP growth ($y_{r,i,t} = \ln(GDP_{r,i,t}^{adj})$) as the dependent variable, with explanatory variables including the lagged GDP growth, the average temperature and precipitation of each season plus fixed-effects for region-industry ($\alpha_{r,i}$) and year-industry ($\alpha_{t,i}$):

$$\Delta y_{r,i,t} = \sum_{s \in S} \beta_{s,i} T_{r,s,t} + \rho_i \Delta y_{r,i,t-1} + \alpha_{r,i} + \alpha_{t,i} + \varepsilon_{r,i,t} \tag{3}$$

where $s$ is the season (either every quarter or every month of the calendar year), $T_{r,s,t}$ is a vector of the weather variables (average temperature, precipitation) affecting the region $r$ in season $s$ of year $t$. We estimate the models by OLS with robust standard-errors clustered by region and year.

In relation to other alternatives such as random-effects, the fixed-effects added in our model help to control for fixed unobservables across time-industry and region-industry without imposing any distribution assumption or any correlation assumption with the other observable variables, while the random-effects models assume that the fixed unobservable errors are normal distributed and uncorrelated with the other observable variables [32, 33]. It is also worth noting that several of the previous papers that estimate the impact of climate change on GDP use fixed-effects rather than random-effects (see [2, 12, 13, 16]).

## Results

### Main regressions

Since the Chilean GDP series for region-industry are available only at an annual frequency, then it is hard to estimate the impact of each month on the yearly GDP of each region-industry (too many coefficients for a 32 year period). However, using only quarterly averages for the weather can mask strong highs and lows in temperature and rainfall. Both the models with monthly weather (too many coefficients and low precision) and quarterly weather (too little

**Table 4. Coefficients for the impact of temperature and precipitation (quarterly averages) on regional industry GDP: OLS with fixed-effects by time and region, separate regressions by industry.**

| | Agriculture | Fishing | Mining | Manufact. | EGA | Constr. | Commerce | Transp. | Finan. serv. | Home | Pers. serv. | Pub. adm. |
|---|---|---|---|---|---|---|---|---|---|---|---|---|
| | (1) | (2) | (3) | (4) | (5) | (6) | (7) | (8) | (9) | (10) | (11) | (12) |
| *Coefficients for Temperature* | | | | | | | | | | | | |
| Jan-Mar | -0.019** | -0.108* | 0.052 | 0.015 | -0.024 | -0.023 | 0.005 | 0.007 | 0.006* | -0.002 | 0.001 | 0.001 |
| | (0.009) | (0.060) | (0.032) | (0.013) | (0.018) | (0.020) | (0.008) | (0.005) | (0.003) | (0.003) | (0.002) | (0.002) |
| Apr-Jun | 0.012 | 0.007 | -0.078** | 0.009 | 0.018* | 0.019 | -0.010 | -0.003 | -0.003 | 0.001 | -0.003 | 0.000 |
| | (0.009) | (0.034) | (0.032) | (0.009) | (0.010) | (0.015) | (0.009) | (0.006) | (0.004) | (0.001) | (0.002) | (0.002) |
| Jul-Sep | -0.012 | 0.034 | 0.077* | -0.008 | 0.018 | 0.011 | 0.005 | 0.000 | 0.005 | -0.003 | 0.003 | -0.002 |
| | (0.011) | (0.039) | (0.043) | (0.013) | (0.018) | (0.016) | (0.004) | (0.007) | (0.004) | (0.003) | (0.003) | (0.002) |
| Oct-Dec | 0.028* | -0.003 | -0.066 | 0.010 | -0.013 | 0.000 | 0.010* | -0.001 | -0.001 | -0.002 | 0.002 | -0.003 |
| | (0.015) | (0.043) | (0.041) | (0.016) | (0.024) | (0.027) | (0.006) | (0.008) | (0.005) | (0.002) | (0.002) | (0.002) |
| *Coefficients for Precipitation* | | | | | | | | | | | | |
| Jan-Mar | 0.005 | -0.001 | 0.017 | 0.010** | 0.007 | -0.003 | -0.001 | -0.002 | -0.000 | -0.000 | 0.001* | -0.001 |
| | (0.004) | (0.012) | (0.018) | (0.005) | (0.008) | (0.010) | (0.002) | (0.002) | (0.001) | (0.000) | (0.001) | (0.001) |
| Apr-Jun | 0.001 | -0.011 | -0.002 | -0.002 | 0.009* | -0.005 | 0.000 | -0.001 | -0.000 | 0.000 | 0.000 | 0.000 |
| | (0.002) | (0.008) | (0.006) | (0.002) | (0.005) | (0.004) | (0.001) | (0.001) | (0.001) | (0.000) | (0.000) | (0.000) |
| Jul-Sep | 0.002 | 0.020** | -0.005 | 0.004 | 0.005 | 0.006 | 0.001 | -0.001 | 0.001 | 0.001 | 0.001 | -0.001 |
| | (0.002) | (0.010) | (0.008) | (0.004) | (0.004) | (0.005) | (0.002) | (0.002) | (0.001) | (0.001) | (0.001) | (0.000) |
| Oct-Dec | 0.000 | -0.002 | -0.011 | -0.007** | 0.007* | 0.009 | -0.000 | -0.006** | -0.001 | -0.000 | 0.000 | -0.000 |
| | (0.004) | (0.008) | (0.010) | (0.003) | (0.004) | (0.007) | (0.002) | (0.003) | (0.001) | (0.001) | (0.001) | (0.001) |
| N | 465 | 436 | 395 | 465 | 460 | 465 | 465 | 465 | 465 | 465 | 465 | 465 |
| $R^2$ | 0.112 | 0.055 | 0.058 | 0.028 | 0.041 | 0.025 | 0.025 | 0.049 | 0.047 | 0.095 | 0.026 | 0.025 |

*Notes*: Observations not weighted for GDP in regressions. Lagged industry GDP growth included in each regression. Robust standard errors clustered by year and region in parentheses.

** $p < 0.01$,

** $p < 0.05$,

* $p < 0.1$

identification from weather shocks) are problematic. Therefore we present both results as alternative models and then comment on their findings.

The quarterly model results (Table 4) only shows a statistically negative impact of temperature on the Agriculture-Silviculture and Fishing sectors. The results by month (Table 5) show a statistically significant negative impact for the temperature of the January month in Agriculture-Silviculture, Fishing, EGA (Electricity, Gas, and Water) and Construction sectors. Therefore our analysis shows that the Agriculture-Silviculture and Fishing sectors in Chile were negatively impacted in a direct way by higher Summer temperatures over the last 35 years, with results being robust for both the monthly and quarterly models. The impact is estimated in terms of reduced-form coefficients, since we cannot verify which channels (such as input-output networks in Chile or value chains effects at the global level) are driving the reduced-form coefficient estimates.

## Robustness checks

The main results are unweighted regressions, with all region-industry pairs with a weight of 1 observation, independently of their economic value. As a robustness check, we repeated the same models with constant weights for each industry and different clustering options (clusters just by year or clusters by region-year). The results were qualitatively similar, although the

**Table 5. Coefficients for the impact of temperature (monthly averages) on regional industry GDP: OLS with fixed-effects by time and region, separate regressions by industry.**

| | Agriculture | Fishing | Mining | Manufact. | EGA | Constr. | Commerce | Transp. | Finan. serv. | Home | Pers. serv. | Pub. adm. |
|---|---|---|---|---|---|---|---|---|---|---|---|---|
| | (1) | (2) | (3) | (4) | (5) | (6) | (7) | (8) | (9) | (10) | (11) | (12) |
| Jan | -0.028*** | -0.121*** | 0.019 | 0.006 | -0.039*** | -0.015* | 0.004 | 0.006 | 0.002 | -0.000 | 0.001 | -0.001 |
| | (0.009) | (0.027) | (0.028) | (0.010) | (0.012) | (0.008) | (0.005) | (0.004) | (0.003) | (0.001) | (0.001) | (0.001) |
| Feb | 0.008 | 0.004 | 0.018** | 0.006 | -0.001 | 0.010 | 0.004* | 0.005 | -0.003 | -0.000 | 0.001 | -0.001 |
| | (0.005) | (0.023) | (0.007) | (0.011) | (0.004) | (0.025) | (0.002) | (0.005) | (0.003) | (0.001) | (0.001) | (0.001) |
| Mar | 0.006 | 0.035 | 0.026 | 0.014 | 0.006 | -0.026 | 0.002 | -0.008 | 0.010*** | -0.004 | 0.000 | 0.003 |
| | (0.010) | (0.034) | (0.030) | (0.010) | (0.016) | (0.018) | (0.006) | (0.006) | (0.003) | (0.002) | (0.002) | (0.002) |
| Apr | 0.008 | -0.042 | -0.020 | -0.015 | 0.023 | 0.009 | -0.011* | 0.005 | 0.000 | 0.000 | -0.005 | -0.000 |
| | (0.006) | (0.050) | (0.025) | (0.010) | (0.016) | (0.023) | (0.006) | (0.003) | (0.004) | (0.001) | (.) | (0.002) |
| May | -0.007* | -0.023 | -0.038* | 0.003 | -0.004 | -0.009 | -0.003 | -0.006 | -0.003 | 0.002 | -0.000 | 0.001 |
| | (0.004) | (0.027) | (0.021) | (0.012) | (0.013) | (0.017) | (0.004) | (0.005) | (0.003) | (0.001) | (0.002) | (0.001) |
| Jun | 0.005 | 0.044 | -0.030*** | 0.009 | 0.004 | 0.017 | -0.001 | -0.001 | -0.001 | -0.001 | 0.000 | -0.000 |
| | (0.005) | (0.034) | (0.009) | (0.011) | (0.010) | (0.019) | (0.005) | (0.003) | (0.003) | (0.001) | (0.001) | (0.001) |
| Jul | 0.004 | 0.007 | 0.055*** | 0.001 | -0.014 | 0.005 | -0.001 | 0.008 | -0.000 | 0.001 | 0.002 | -0.001 |
| | (0.008) | (0.023) | (0.017) | (0.008) | (0.019) | (0.019) | (0.005) | (0.005) | (0.003) | (0.001) | (0.002) | (0.002) |
| Aug | -0.000 | 0.019 | -0.018 | 0.009 | 0.037 | 0.047*** | 0.007 | -0.006 | -0.000 | -0.002 | -0.001 | 0.001 |
| | (0.009) | (0.023) | (.) | (0.019) | (0.028) | (0.018) | (0.004) | (0.006) | (0.003) | (0.003) | (0.002) | (0.001) |
| Sep | -0.018*** | -0.049 | 0.036 | -0.022** | 0.001 | -0.035** | -0.000 | 0.002 | 0.007* | -0.001 | 0.001 | -0.002 |
| | (0.007) | (0.049) | (0.024) | (0.009) | (0.018) | (0.017) | (0.006) | (0.002) | (0.004) | (0.002) | (0.002) | (0.002) |
| Oct | 0.012** | 0.047 | -0.003 | 0.003 | 0.007 | -0.016 | 0.002 | -0.003 | -0.006 | -0.001 | 0.002* | -0.000 |
| | (0.005) | (0.042) | (0.020) | (0.009) | (0.017) | (0.021) | (0.005) | (0.005) | (0.005) | (0.002) | (0.001) | (0.002) |
| Nov | 0.023 | -0.010 | -0.027 | 0.009 | -0.020 | -0.003 | 0.001 | 0.001 | 0.005 | -0.001 | 0.001 | -0.002 |
| | (0.015) | (0.042) | (0.027) | (0.016) | (0.016) | (0.028) | (0.006) | (0.003) | (0.005) | (0.001) | (0.001) | (0.002) |
| Dec | 0.000 | 0.001 | -0.037 | 0.002 | 0.019 | 0.028 | 0.006* | -0.001 | -0.000 | -0.000 | -0.001 | 0.000 |
| | (0.006) | (0.027) | (0.029) | (0.016) | (.) | (0.026) | (0.003) | (0.005) | (0.002) | (0.001) | (0.001) | (0.001) |
| $N$ | 465 | 436 | 395 | 465 | 460 | 465 | 465 | 465 | 465 | 465 | 465 | 465 |
| $R^2$ | 0.187 | 0.113 | 0.087 | 0.065 | 0.112 | 0.082 | 0.058 | 0.087 | 0.106 | 0.158 | 0.060 | 0.054 |

*Notes*: Observations not weighted for GDP in regressions. Lagged industry growth rate and monthly precipitation included in regressions. Robust standard errors clustered by year and region in parentheses.

*** $p < 0.01$,

** $p < 0.05$,

* $p < 0.1$

coefficients for Fishing lost statistical significance (see Table B1 in S2 Appendix). We also include an exercise that aggregates regions I and XV plus regions X and XIV, therefore presenting 13 regions for the entire period of 1985 to 2017 (see Table B2 in S2 Appendix with the quarterly temperatures' model and Table B1 in S2 Appendix with the monthly temperatures' model).

## Calibrated projections of the climate change impact for 2050 and 2100

Now we use the estimated coefficients from the model with the monthly weather fluctuations (Table 5) to implement a calibrated exercise using the global temperature projections of the IPCC (2014) [14] to project how a uniform temperature increase throughout the entire year may affect the Chilean GDP. Climate studies consider several scenarios given by Representative Concentration Pathways (RCP), with RCP 2.6 being denoted as the best possible scenario

in which climate change is completely controlled, RCP 4.5 being a scenario in which the global temperature rise is likely to fall below 2.0, and RCP 8.5 being considered the worst scenario in which no country implements policies or mitigators for climate change [14].

The quantitative exercise considers the impact of a given global temperature change in climate change, $T_t^{RCP-x}$, in year $t$ for each $RCP - x$ path (with $x$ = 2.6, 4.5, 6.0, 8.5) on the GDP growth rate and on the GDP level of each industry $i$: $I - growth_{i,t} = (\sum_{s=1}^{12} \beta_{s,i})(T_t^{RCP-x} - T_{2017})$ and $I - level_{i,t} = \exp(\sum_{t'=2017}^{t} I - growth_{i,t'}) - 1$. These RCP paths scenarios are obtained from the average path values of the United Nations modeling experts [14]. These path values are widely used in macro-financial stress tests with climate change factors [34]. We then obtain the estimates of the impact on the aggregate GDP by summing up across all industries, $I - growth_t = \sum_{i=1}^{12} w_{i,t} I - growth_{i,t}$ and $I - level_t = \sum_{i=1}^{12} w_{i,t} I - level_{i,t}$, with $w_{i,t}$ denoting the weight of each industry $i$ in the total GDP at time $t$. Notice that the GDP growth rate and level impact costs are measured relative to a world with no climate change and not relative to 2017.

We produce 3 estimates of the impact of climate change on each industry $i$ at horizon $t$: 1) using all the model's estimated point coefficients for the effect of the temperature on the industrial GDP ($\beta = (\hat{\beta}_{1,1}, .., \hat{\beta}_{s,i}, .., \hat{\beta}_{12,12})$ with $s$ = 1,.., 12 denoting month and $i$ = 1, . . ., 12 denoting industry; 2) using only the statistically significant coefficients at a level of 10% or lower ($\beta = (.., \beta_{s,i}..)$ with $\beta_{s,i} = \hat{\beta}_{s,i} 1\left(\frac{|\hat{\beta}_{s,i}|}{SE(\hat{\beta}_{s,i})} \leq 1.65\right)$, with $1(.)$ being the indicator function); and 3) using only negative coefficients (that is, disregarding positive impacts of climate change) that are statistically significant ($\beta = (.., \beta_{s,i}..)$ with $\beta_{s,i} = \hat{\beta}_{s,i} 1\left(\frac{|\hat{\beta}_{s,i}|}{SE(\hat{\beta}_{s,i})} \leq 1.65\right) 1(\hat{\beta}_{s,i} < 0)$).

Table 6 denotes the values of the estimated sum of the coefficients for all months for each industry $i$ ($\sum_{s=1}^{12} \beta_{s,i}$) under each of these separate assumptions, which represents the impact on the GDP growth rate of each industry in a given year for a uniform increase in temperature of 1 C throughout all the months of the year. If one considers the quarterly weather model from Table 4, then Fishing and Mining are the only industries which are negatively impacted by climate change with a statistical significance, although the coefficient for Mining is very small. By considering the monthly fluctuations model from Table 5, then we find a statistically significant impact of an increase of 1 C in temperature that reduces the growth rate in Agriculture, Fishing, Manufacture, EGA (Electricity, Gas, and Water), Construction and Commerce of -3%, -12.1%, -2.2%, -3.9%, -0.3% and -0.7%, respectively.

**Table 6. Total impact on the industry GDP growth rate (in %) of the estimated models for a one degree Celsius temperature increase throughout the year.**

| Sum of the coefficients' impact | Industries | | | | | | | | | | | |
|---|---|---|---|---|---|---|---|---|---|---|---|---|
| | (1) | (2) | (3) | (4) | (5) | (6) | (7) | (8) | (9) | (10) | (11) | (12) |
| *Quarterly weather fluctuations model* (Table 4) | | | | | | | | | | | | |
| All quarters | 0.9 | -7.0 | -1.5 | 2.6 | -0.1 | 0.7 | 1.0 | 0.3 | 0.7 | -0.6 | 0.3 | -0.4 |
| Statistically significant | 0.9 | -10.8 | -0.1 | 0.0 | 1.8 | 0.0 | 1.0 | 0.0 | 0.6 | 0.0 | 0.0 | 0.0 |
| Significant & negative | 0.0 | -10.8 | -0.1 | 0.0 | 0.0 | 0.0 | 0.0 | 0.0 | 0.0 | 0.0 | 0.0 | 0.0 |
| *Monthly weather fluctuations model* (Table 5) | | | | | | | | | | | | |
| All months | 1.3 | -8.8 | -1.9 | 2.5 | 1.9 | 1.2 | 0.4 | 0.2 | 1.1 | -0.7 | 0.1 | -0.2 |
| Statistically significant | -3.0 | -12.1 | 0.5 | -2.2 | -3.9 | -0.3 | -0.7 | 0.0 | 1.7 | 0.0 | 0.2 | 0.0 |
| Significant & negative | -3.0 | -12.1 | 0.0 | -2.2 | -3.9 | -0.3 | -0.7 | 0.0 | 0.0 | 0.0 | 0.0 | 0.0 |

*Notes*: (1) Agriculture and Forestry, (2) Fishing, (3) Mining, (4) Manufacturing, (5) EGA, (6) Construction, (7) Commerce, Restaurants, and Hotels, (8) Transport and Communications, (9) Financial Services, (10) Home Ownership, (11) Personal Services, (12) Public Administration.

**Table 7. Simulated impact (in %) of the climate change on the industry, overall GDP and growth rates in Chile for 1985–2017 and for future (monthly, all coefficients, Table 5).**

| Temperature increase | Industries | | | | | | | | | | | | Total GDP |
|---|---|---|---|---|---|---|---|---|---|---|---|---|---|
| | **(1)** | **(2)** | **(3)** | **(4)** | **(5)** | **(6)** | **(7)** | **(8)** | **(9)** | **(10)** | **(11)** | **(12)** | |
| *Impact on GDP growth rate in 2017 relative to no warming after 1985* | | | | | | | | | | | | | |
| 0.26˚C* | 0.3 | -2.3 | -0.5 | 0.6 | 0.5 | 0.3 | 0.1 | 0.1 | 0.3 | -0.2 | 0.0 | -0.1 | 0.1 |
| *Impact on GDP level in 2050 relative to no warming after 2017* | | | | | | | | | | | | | |
| 1.0˚C | 24.7 | -77.6 | -27.6 | 53.0 | 38.1 | 22.6 | 7.0 | 3.5 | 20.6 | -11.2 | 1.7 | -3.3 | 9.8 |
| 1.3˚C | 33.3 | -85.7 | -34.3 | 73.8 | 52.2 | 30.4 | 9.2 | 4.5 | 27.5 | -14.3 | 2.2 | -4.3 | 14.0 |
| 1.4˚C | 36.3 | -87.7 | -36.4 | 81.3 | 57.2 | 33.1 | 10.0 | 4.9 | 29.9 | -15.3 | 2.4 | -4.6 | 15.5 |
| 2.0˚C | 55.6 | -95.0 | -47.6 | 134.0 | 90.8 | 50.4 | 14.6 | 7.0 | 45.4 | -21.2 | 3.5 | -6.6 | 26.0 |
| *Impact on GDP level in 2100 relative to no warming after 2017* | | | | | | | | | | | | | |
| 1.0˚C | 72.6 | -97.5 | -55.0 | 185.8 | 122.1 | 65.5 | 18.3 | 8.8 | 58.7 | -25.5 | 4.3 | -8.1 | 36.3 |
| 1.8˚C | 167.2 | -99.9 | -76.2 | 561.9 | 320.6 | 147.7 | 35.3 | 16.3 | 129.7 | -41.1 | 7.9 | -14.0 | 105.6 |
| 2.2˚C | 232.4 | -100 | -82.7 | 907.4 | 478.7 | 203.1 | 44.7 | 20.3 | 176.3 | -47.6 | 9.7 | -16.9 | 164.4 |
| 3.7˚C | 654.0 | -100 | -94.8 | 4766.7 | 1815.6 | 545.5 | 86.2 | 36.5 | 452.6 | -66.3 | 16.8 | -26.7 | 738.0 |

*Notes*:

*Global temperature increase over the period 1985–2017.

(1) Agriculture and Forestry, (2) Fishing, (3) Mining, (4) Manufacturing, (5) EGA, (6) Construction, (7) Commerce, Restaurants, and Hotels, (8) Transport and Communications, (9) Financial Services, (10) Home Ownership, (11) Personal Services, (12) Public Administration.

Table 7 shows the impact of the average global temperature increase according to different climate emission paths [14] under the assumption that we apply all the model's coefficients in the forecast. Under this assumption, Fishing, Mining, Home property and Public administration are the only industries hurt by climate change whether at the horizons of 2050 or 2100. In particular, Fishing's GDP almost disappears by 2100, even with just a 1.0 C increase in temperature. With all the model's estimated coefficients, Mining and Home property would also decrease by at least 55% and 25.5%, respectively, by 2100. However, climate change would have a strong positive impact on the other economic sectors and therefore the total Chilean GDP would increase across all scenarios in 2050 and 2100. It is unlikely that such large positive impacts of climate change may materialize, however, since these projections obviously assume that the coefficients are fixed over time. Probably it is more realistic to assume that the positive impacts of climate change may decline over time and even turn into negative effects.

Table 8 shows the impact of the average global temperature increase according to different climate emission paths [14] under the assumption that we apply only the model's coefficients that are statistically significant at the 10% level at least. Under this assumption, Agriculture, Fishing, Manufacture, EGA, Construction and Commerce are the only industries hurt by climate change whether at the horizons of 2050 or 2100. Even with just a 1.0 C increase in temperature, Agriculture, Fishing, Manufacture, EGA, Construction and Commerce would decline by 71.6%, 99.4%, 60.3%, 80.6%, 11.8% and 25.5%, respectively, around 2100. However, climate change would have a strong positive impact on the other economic sectors and therefore the total Chilean GDP would change only slightly in 2050 and it would even increase across all scenarios in 2100. Again, however, this result is strongly dependent on the positive effects of climate change estimated for some sectors and these effects may not materialize, since such positive effects may decline over time and even turn into negative effects.

Finally, Table 9 shows the impact of the average global temperature increase according to different climate emission paths [14] using only the model's coefficients that are both negative

**Table 8. Simulated impact of the climate change on the industry, overall GDP and growth rates in Chile: Only stat. significant coefficients (monthly, Table 5).**

| Temperature increase | Industries | | | | | | | | | | | | Total GDP |
|---|---|---|---|---|---|---|---|---|---|---|---|---|---|
| | (1) | (2) | (3) | (4) | (5) | (6) | (7) | (8) | (9) | (10) | (11) | (12) | |
| *Impact on GDP growth rate in 2017 relative to no warming after 1985* | | | | | | | | | | | | | |
| 0.26˚C* | -0.8 | -3.1 | 0.1 | -0.6 | -1.0 | -0.1 | -0.2 | 0 | 0.4 | 0 | 0.1 | 0 | -0.1 |
| *Impact on GDP level in 2050 relative to no warming after 2017**\*** | | | | | | | | | | | | | |
| 1.0˚C | -40.0 | -87.2 | 8.9 | -31.2 | -48.5 | -5.0 | -11.2 | 0 | 33.5 | 0 | 3.5 | 0 | -2.3 |
| 1.3˚C | -48.5 | -93.1 | 11.7 | -38.5 | -57.8 | -6.4 | -14.3 | 0 | 45.6 | 0 | 4.5 | 0 | -1.9 |
| 1.4˚C | -51.0 | -94.4 | 12.6 | -40.8 | -60.5 | -6.9 | -15.3 | 0 | 49.9 | 0 | 4.9 | 0 | -1.7 |
| 2.0˚C | -63.9 | -98.4 | 18.5 | -52.7 | -73.4 | -9.7 | -21.2 | 0 | 78.2 | 0 | 7.0 | 0 | 0.4 |
| *Impact on GDP level in 2100 relative to no warming after 2017**\*** | | | | | | | | | | | | | |
| 1.0˚C | -71.6 | -99.4 | 23.4 | -60.3 | -80.6 | -11.8 | -25.5 | 0 | 104.2 | 0 | 8.8 | 0 | 3.2 |
| 1.8˚C | -89.6 | -100 | 45.9 | -81.0 | -94.8 | -20.3 | -41.1 | 0 | 261.5 | 0 | 16.3 | 0 | 25.3 |
| 2.2˚C | -93.7 | -100 | 58.7 | -86.9 | -97.3 | -24.2 | -47.6 | 0 | 381.0 | 0 | 20.3 | 0 | 44.0 |
| 3.7˚C | -99.1 | -100 | 117.5 | -96.7 | -99.8 | -37.3 | -66.3 | 0 | 1303.8 | 0 | 36.5 | 0 | 193.2 |

*Notes*:

*Global temperature increase over the period 1985–2017.

**RCP 2.6, 4.5, 6.0, 8.5.

(1) Agriculture and Forestry, (2) Fishing, (3) Mining, (4) Manufacturing, (5) EGA, (6) Construction, (7) Commerce, Restaurants, and Hotels, (8) Transport and Communications, (9) Financial Services, (10) Home Ownership, (11) Personal Services, (12) Public Administration.

and statistically significant. Again, under this assumption, Agriculture, Fishing, Manufacture, EGA, Construction and Commerce are the only industries hurt by climate change whether at the horizons of 2050 or 2100. In terms of the negative impact of climate change on the total GDP, it could range between 9% and 14.8% in 2050 and between 16.8% and 29.6% in 2100.

**Table 9. Simulated impact of the climate change on the industry, overall GDP and growth rates in Chile: Only stat. significant coefficients with a negative value (monthly, Table 5).**

| Temperature increase | Industries | | | | | | | | | | | | Total GDP |
|---|---|---|---|---|---|---|---|---|---|---|---|---|---|
| | (1) | (2) | (3) | (4) | (5) | (6) | (7) | (8) | (9) | (10) | (11) | (12) | |
| *Impact on GDP growth rate in 2017 relative to no warming after 1985**\*** | | | | | | | | | | | | | |
| 0.26˚C* | -0.8 | -3.1 | 0 | -0.6 | -1.0 | -0.1 | -0.2 | 0 | 0 | 0 | 0 | 0 | -0.2 |
| *Impact on GDP level in 2050 relative to no warming after 2017**\*** | | | | | | | | | | | | | |
| 1.0˚C | -40.0 | -87.2 | 0 | -31.2 | -48.5 | -5.0 | -11.2 | 0 | 0 | 0 | 0 | 0 | -9.0 |
| 1.3˚C | -48.5 | -93.1 | 0 | -38.5 | -57.8 | -6.4 | -14.3 | 0 | 0 | 0 | 0 | 0 | -10.9 |
| 1.4˚C | -51.0 | -94.4 | 0 | -40.8 | -60.5 | -6.9 | -15.3 | 0 | 0 | 0 | 0 | 0 | -11.5 |
| 2.0˚C | -63.9 | -98.4 | 0 | -52.7 | -73.4 | -9.7 | -21.2 | 0 | 0 | 0 | 0 | 0 | -14.8 |
| *Impact on GDP level in 2100 relative to no warming after 2017**\*** | | | | | | | | | | | | | |
| 1.0˚C | -71.6 | -99.4 | 0 | -60.3 | -80.6 | -11.8 | -25.5 | 0 | 0 | 0 | 0 | 0 | -16.8 |
| 1.8˚C | -89.6 | -100 | 0 | -81.0 | -94.8 | -20.3 | -41.1 | 0 | 0 | 0 | 0 | 0 | -22.9 |
| 2.2˚C | -93.7 | -100 | 0 | -86.9 | -97.3 | -24.2 | -47.6 | 0 | 0 | 0 | 0 | 0 | -24.9 |
| 3.7˚C | -99.1 | -100 | 0 | -96.7 | -99.8 | -37.3 | -66.3 | 0 | 0 | 0 | 0 | 0 | -29.6 |

*Notes*:

*Global temperature increase over the period 1985–2017.

**RCP 2.6, 4.5, 6.0, 8.5.

(1) Agriculture and Forestry, (2) Fishing, (3) Mining, (4) Manufacturing, (5) EGA, (6) Construction, (7) Commerce, Restaurants, and Hotels, (8) Transport and Communications, (9) Financial Services, (10) Home Ownership, (11) Personal Services, (12) Public Administration.

Therefore our model predicts a large and positive impact of climate change on the total Chilean GDP if one uses all the model's coefficients both in 2050 and 2100, a small impact of climate change in 2050 if the forecasts use just the statistically significant coefficients, and a moderately negative impact of climate change both in 2050 and 2100 if the forecasts apply just the negative and statistically significant coefficients. It is possible that the forecasts using all the model's coefficients are way too optimistic, while the forecasts with just the negative and statistically significant coefficients can be too pessimistic since the coefficients are selected to clearly present a negative scenario. The appendix at the end of the article shows a counterfactual exercise with the most recent temperature projection paths of the IPCC (2021) [15], but the results are broadly similar, both qualitatively and quantitatively.

## Calibrated projections for Chile using industry coefficients estimated for the USA

The counterfactual exercises considered in Tables 7–9 implemented the coefficients estimated from our model in Table 5. However, Chile only has 15 regions and many of those regions have a zero value for some industries or an extremely low value. The small number of Chilean regions and the low economic value of several of those regions makes it harder to estimate the economic impact of climate change in a reliable way. For this reason, we also implement a counterfactual exercise where the impact coefficient of the temperature increase induced by climate change is obtained from a model estimated by Colacito, Hoffmann and Phan [13] for the 50 states and 12 industries of the USA. Since the USA' states are relatively large economies and it includes a large number of states, then the impact of climate change on each industry can be estimated in a more precise way and with lower standard errors. The difference is that we are applying the value of each Chilean industry on GDP to make the projection for total GDP and this considers that Chile has higher shares of some industries such as Mining or Agriculture and lower shares of other industries. One small note is that we apply the same US coefficients for Agriculture-Fishing and Finance-Insurance-Real Estate to the separate Chilean industries of Agriculture and Fishing plus Financial services and Home property, since those industries are treated separately in the Chilean national accounts data.

Table 10 shows the counterfactual exercise of applying the coefficients from **Table A21** in Colacito, Hoffmann and Phan (2019) [13]. It shows its strongest impact in 2050 on Construction, which would decline between 12.7% and 23.8%. Home property, Financial Services and Manufactures would also be strongly hit, declining between 9% and 19.5% relative to a scenario with no climate change. However, due to the positive coefficient estimated for the Mining industry, the impact of climate change on the Chilean GDP in 2050 would be limited to a decline of 0.8% or less. In 2100 the projections for climate change's impact on total GDP would again become very positive, because the counterfactual would assume that the log-growth of Mining would add up linearly over time.

Table 11 shows a very similar counterfactual exercise with impact coefficients for temperature estimated for the US, but it considers only the statistically significant coefficients from **Table A21** in Colacito, Hoffmann and Phan (2019) [13]. The exercise also applies the Agriculture-Fishing and Manufacturing industries coefficients from **Table A20** in Colacito, Hoffmann and Phan (2019) [13], since those coefficients were estimated with a smaller standard-error, perhaps due to the higher importance of such industries for the US economy before 1997. The results show a negative impact of climate change for most industries, except for Mining, Energy-Gas-Water (EGA) and Transports-Communications. The strongest impact of climate change is now estimated to be for the Agriculture and Fishing sectors, followed by Construction, Financial services and Home property. In particular, Agriculture and Fishing

**Table 10. Simulated impact (in %) of the climate change on the industry and overall GDP level and growth rates in Chile for the future[*].**

| Temperature increase | Industries | | | | | | | | | | | | Total GDP |
|---|---|---|---|---|---|---|---|---|---|---|---|---|---|
| | (1) | (2) | (3) | (4) | (5) | (6) | (7) | (8) | (9) | (10) | (11) | (12) | |
| *Impact on GDP level in 2050 relative to no warming after 2017* | | | | | | | | | | | | | |
| 1.0˚C | -4.6 | -4.6 | 53.4 | -9.1 | 7.0 | -12.7 | -6.2 | 0.2 | -10.3 | -10.3 | -7.7 | -5.6 | -0.8 |
| 1.3˚C | -5.9 | -5.9 | 74.4 | -11.7 | 9.1 | -16.2 | -8.0 | 0.3 | -13.2 | -13.2 | -9.9 | -7.2 | -0.4 |
| 1.4˚C | -6.3 | -6.3 | 82.0 | -12.5 | 9.9 | -17.3 | -8.6 | 0.3 | -14.1 | -14.1 | -10.6 | -7.7 | -0.2 |
| 2.0˚C | -8.9 | -8.9 | 135.2 | -17.4 | 14.4 | -23.8 | -12.0 | 0.4 | -19.5 | -19.5 | -14.8 | -10.9 | 2.0 |
| *Impact on GDP level in 2100 relative to no warming after 2017* | | | | | | | | | | | | | |
| 1.0˚C | -10.9 | -10.9 | 187.7 | -21.0 | 18.1 | -28.5 | -14.6 | 0.5 | -23.5 | -23.5 | -18.0 | -13.2 | 5.0 |
| 1.8˚C | -18.7 | -18.7 | 570.0 | -34.6 | 34.9 | -45.4 | -24.8 | 1.0 | -38.3 | -38.3 | -30.0 | -22.5 | 36.1 |
| 2.2˚C | -22.4 | -22.4 | 922.4 | -40.5 | 44.2 | -52.3 | -29.4 | 1.2 | -44.5 | -44.5 | -35.3 | -26.8 | 69.4 |
| 3.7˚C | -34.7 | -34.7 | 4889.2 | -58.2 | 85.0 | -71.2 | -44.3 | 2.0 | -62.9 | -62.9 | -52.0 | -40.9 | 477.6 |

*Notes*:

[*]Climate change temperature coefficients estimated for the USA industry (post-1997) from Table A21 (both) years in Colacito, Hoffmann and Phan (2019).

(1) Agriculture and Forestry, (2) Fishing, (3) Mining, (4) Manufacturing, (5) EGA, (6) Construction, (7) Commerce, Restaurants, and Hotels, (8) Transport and Communications, (9) Financial Services, (10) Home Ownership, (11) Personal Services, (12) Public Administration.

may decline between 18.7% and 33.9% in 2050, relative to a scenario with no additional climate change. Construction, Financial services and Home property would decline between 10.3% and 23.8% in 2050 due to the impact of worsening climate change. In terms of the total GDP, the effect of climate change in 2050 would imply a deterioration between 6.8% and 12.9%. By 2100 the Agriculture and Fishing industries would decline between 40% and 84.9%, while the Construction, Financial services and Home property would decline between 23.5% and 71.2% due to climate change. Climate change by 2100 could imply a deterioration between 15.5% and 42.3% on the total Chilean GDP.

**Table 11. Simulated impact (in %) of the climate change on the industry and overall GDP level and growth rates in Chile for the future[*].**

| Temperature increase | Industries | | | | | | | | | | | | Total GDP |
|---|---|---|---|---|---|---|---|---|---|---|---|---|---|
| | (1) | (2) | (3) | (4) | (5) | (6) | (7) | (8) | (9) | (10) | (11) | (12) | |
| *Impact on GDP level in 2050 relative to no warming after 2017* | | | | | | | | | | | | | |
| 1.0˚C | -18.7 | -18.7 | 0.0 | -4.6 | 0.0 | -12.7 | -6.2 | 0.0 | -10.3 | -10.3 | -7.7 | -5.6 | -6.8 |
| 1.3˚C | -23.6 | -23.6 | 0.0 | -5.9 | 0.0 | -16.2 | -8.0 | 0.0 | -13.2 | -13.2 | -9.9 | -7.2 | -8.7 |
| 1.4˚C | -25.2 | -25.2 | 0.0 | -6.3 | 0.0 | -17.3 | -8.6 | 0.0 | -14.1 | -14.1 | -10.6 | -7.7 | -9.3 |
| 2.0˚C | -33.9 | -33.9 | 0.0 | -8.9 | 0.0 | -23.8 | -12.0 | 0.0 | -19.5 | -19.5 | -14.8 | -10.9 | -12.9 |
| *Impact on GDP level in 2100 relative to no warming after 2017* | | | | | | | | | | | | | |
| 1.0˚C | -40.0 | -40.0 | 0.0 | -10.9 | 0.0 | -28.5 | -14.6 | 0.0 | -23.5 | -23.5 | -18.0 | -13.2 | -15.5 |
| 1.8˚C | -60.2 | -60.2 | 0.0 | -18.7 | 0.0 | -45.4 | -24.8 | 0.0 | -38.3 | -38.3 | -30.0 | -22.5 | -25.4 |
| 2.2˚C | -67.5 | -67.5 | 0.0 | -22.4 | 0.0 | -52.3 | -29.4 | 0.0 | -44.5 | -44.5 | -35.3 | -26.8 | -29.6 |
| 3.7˚C | -84.9 | -84.9 | 0.0 | -34.7 | 0.0 | -71.2 | -44.3 | 0.0 | -62.9 | -62.9 | -52.0 | -40.9 | -42.3 |

*Notes*:

[*]Climate change temperature coefficients estimated for the USA industry (post-1997) from Table A21 plus Agriculture- (both years, only statistically significant coefficients) Fishing and Manufacturing industries coefficients from Table A20 (pre-1997) in Colacito, Hoffmann and Phan (2019).

(1) Agriculture and Forestry, (2) Fishing, (3) Mining, (4) Manufacturing, (5) EGA, (6) Construction, (7) Commerce, Restaurants, and Hotels, (8) Transport and Communications, (9) Financial Services, (10) Home Ownership, (11) Personal Services, (12) Public Administration

## Conclusions and policy implications

Based on annual region-industry panel data for the period 1985 to 2017, our study finds that climate change had little effect on the different sectors of economic activity in Chile over the last 35 years. We found a statistically significant negative effect of climate change in Chile, with the channel coming from higher temperatures rather than fluctuations in precipitation. We find that high temperatures in the summer season (January to March) had a negative impact on the Agriculture-Silviculture and Fishing sectors. Furthermore, by separating the weather at a monthly level, we find that it is high temperature in January in particular, which causes the strongest negative impact. Higher temperatures in January may also cause some deterioration of activity for the Construction and EGA (Electricity, Gas, and Water) sectors. However, since Agriculture-Silviculture and Fishing represent just 4% of GDP and summing the sectors of Construction (6.8% of GDP) and Electricity, Gas, and Water (4.0% of GDP), the analysis shows that 85% of the economic activity in Chile was not affected by climate change and that such effect was limited to either the summer season (January to March) or even just a single month (January).

We also find that higher temperatures in some seasons, such as the Spring, can have a positive impact on economic growth, which confirms previous results found for the USA [13]. For instance, Agriculture is positively affected by the temperature increases during the month of November. If we consider the point estimates for all the model coefficients across every month (whether the coefficients are statistically significant or not), then each additional Celsius degree of temperature decreases GDP by -8.8% in Fishing, -1.9% in Mining and -0.7% in the Home property sector, but it shows a positive impact on the output of other sectors, including Agriculture. If one considers just the statistically significant coefficients, then each Celsius degree of temperature decreases GDP by -3% in Agriculture, -8.8% in Fishing, -2.2% in Manufacturing, -3.9% in Energy-Gas-Water, -0.3% in Construction and -0.7% in Commerce, but it still has a positive impact on several other sectors such as Mining, Finance and Personal Services. Unfortunately, extreme weather is often associated with a single month or even shorter periods, therefore the unavailability of regional-industry GDP data at a quarterly or monthly frequency makes statistical identification harder and casts some uncertainty on the interpretation of our findings.

We then use our model to present several projections of the impact of climate change on the GDP of each industry and the total national GDP by 2050 and 2100. These projections consider the average of the global climate paths published by the United Nations [14, 15], which are widely used in climate stress tests [34]. Over time, the fraction of GDP represented by the sectors economically affected by climate change falls, with Agriculture and Fishing almost disappearing in terms of their weight on the GDP, and this limits the negative impact of climate change on GDP even as global temperatures become worse. The stress test exercises are robust to using either the 2014 or the 2021 scenarios of the IPCC.

These projections are very sensitive to whether we consider all the model's coefficients, only the statistically significant coefficients, or just the negative statistically significant coefficients (that is, ignoring potential positive effects of climate change). Considering all the model's coefficients we obtain a large and positive impact of climate change on the Chilean GDP level, with a range between +9.8% and +26.% in 2050 and between +36% and 738% by 2100. Note, however, that this positive impact of climate change depends on statistically insignificant coefficients and also on fixed coefficients that do not consider that its effects may change over time. Using only the model's statistically significant coefficients, we obtain an impact on the Chilean GDP level between -2.3% and +0.4% in 2050 and between +3.2% and +193.2% in 2100. The reason why some increases in temperature can cause increases in GDP is because

the sectors that benefit from climate change increase their weight in the Chilean economy, while the negatively affected sectors cannot decrease their product below a GDP level of zero. In our worst forecasts, which apply only negative coefficients that are also statistically significant (that is, ignoring any potential positive effects), we then obtain an impact on the Chilean GDP level between -14.8% and -9% in 2050 and between -29.6% and -16.8% in 2100. That is, we only obtain a negative impact of climate change on the total Chilean GDP if we deliberately ignore any positive coefficients.

A robustness exercise using the industry coefficients of a similar model estimated for the USA [13] would imply that Chile's GDP would suffer a fall of at most 0.8% by 2050 and would increase substantially by 2100 due to the effects of climate change. However, a second robustness exercise that applies only the statistically significant coefficients estimated for the USA [13] would imply that the Chilean GDP would fall between -6.8% and -12.9% in 2050 and between -15.5% and -42.3% in 2100 due to climate change.

Our estimates also imply a positive impact on the Chilean growth rate during the period 1985–2017 of +0.1% with all the model's coefficients, a negative effect of -0.1% with just the statistically significant coefficients, and a negative effect of -0.2% with just the negative statistically significant coefficients. Therefore it does not appear that climate change had an impact for Chilean economic activity in the past. Although there are other caveats, one that is directly related to the estimation is that the annual frequency of the region-industry data makes it harder to measure the impact of climate change associated with just one month. Another issue that causes uncertainty in our results is that our model has fixed coefficients instead of time-varying parameters that consider the dynamic impacts of climate change over time. Neither issue can be solved in a model based only on an annual frequency panel dataset.

One policy implications of this work is that more research is required for knowing whether the effects of climate change are permanent over the long term or not. Our work finds an impact of seasonal temperatures on the growth rate of the GDP of several industries and an effect on the growth rate may have large accumulated impacts over several years, as shown in our exercises for Chile and previous studies for the USA [13]. However, many industries may undertake investments to mitigate the effects of climate change, such as finding alternative energy sources or crops that are better suited to warmer weather [35]. Governments can also implement new regulations and build better infrastructure to adjust for the long run climate. Recent research, however, has found evidence of fairly negative effects of climate change on agriculture even after several decades [29, 36], showing that current adjustments may not be enough to mitigate the negative shock of global warming. It is therefore crucial for economic research to provide greater evidence on all the possible short-run and long-run effects of climate change on different industries and natural resources [37] in order to evaluate the value of environmental regulations and green investments [23].

## Supporting information

**S1 Appendix. Panel-level heterogeneity and unit root tests.**
(PDF)

**S2 Appendix. Other model estimates.** This appendix show some robustness checks using the same model of industry-region GDP with temperature and precipitation fluctuations with constant weights for each industry and different clustering options (clusters just by year or clusters by region-year).
(PDF)

**S3 Appendix. Calibrated projections of climate change for Chile using the new IPCC (2021) SSP scenarios.** This appendix considers counterfactual exercises using the most recent "Shared Socioeconomic Pathways" (SSPs) scenarios published by the IPCC's Sixth Assessment Report (IPCC 2021).
(PDF)

**S4 Appendix. Precipitation and temperature evolution statistics between 1950 and 2017.** This appendix shows the results of the yearly temperature and precipitation fluctuations by macrozone weighted by the GDP of each region.
(PDF)

**S5 Appendix. GDP across regions.**
(PDF)

## Acknowledgments

We thank Bridget Hoffmann, Toàn Phan and seminar participants at the Central Bank of Chile for comments. The views expressed in this work do not represent the Central Bank of Chile. All errors are our own.

## Author Contributions

**Conceptualization:** Carlos Madeira.

**Data curation:** Karla Hernandez, Carlos Madeira.

**Formal analysis:** Carlos Madeira.

**Investigation:** Karla Hernandez, Carlos Madeira.

**Methodology:** Carlos Madeira.

**Project administration:** Carlos Madeira.

**Resources:** Carlos Madeira.

**Software:** Karla Hernandez, Carlos Madeira.

**Supervision:** Carlos Madeira.

**Validation:** Carlos Madeira.

**Visualization:** Carlos Madeira.

**Writing – original draft:** Carlos Madeira.

**Writing – review & editing:** Carlos Madeira.

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
