## [Decision Letter · Decision Letter 0]

16 Dec 2021

PONE-D-21-33259The impact of climate change on economic output across industries in ChilePLOS ONE

Dear Dr. Madeira,

Thank you for submitting your manuscript to PLOS ONE. After careful consideration, we feel that it has merit but does not fully meet PLOS ONE’s publication criteria as it currently stands. Therefore, we invite you to submit a revised version of the manuscript that addresses the points raised during the review process.

We look forward to receiving your revised manuscript.

Kind regards,

Francisco X Aguilar

Academic Editor

PLOS ONE

Journal Requirements:

Additional Editor Comments:

Two reviewers have offered constructive criticisms to the submitted manuscript, all of which shall be fully addressed. For instance, the authors should clearly state the contribution of this study and the rationale supporting their model specification. Within their econometric estimation, a justification for not using common approaches to panel data (e.g. fixed, random, mixed effects) is necessary. Preferably, and if relevant, such models should be run. The model seems to be later calibrated with US-based data, which should be properly motivated and explained within a Methods section.

Editorial observations: Section 2 should be re-titled to 'Methods' or 'Methods and Data'. Other subtitles such as 'Results' should be simply labeled as such (the Results are for Chile as clearly stated from the Title, so there is no need to include 'Chile' in a subtitle). The language is adequate but some careful editing is needed. As a case in point, the authors write: "The analysis for the past 35 years would show that 85% of the economic activity..." In this case "past" should be avoided as there will likely be a mismatch between time of publication and the period covered in the study. The word "would" is not needed, as their econometrics results indeed show these trends. A revised text could read" Our analysis over the 1985-2017 period show that 85% of the economic activity...". Although this might seem trivial, it will help with readability and possible increase the impact of the manuscript.

Reviewers' comments:

Reviewer's Responses to Questions

**Comments to the Author**

1. Is the manuscript technically sound, and do the data support the conclusions?

Reviewer #1: Partly

Reviewer #2: Yes

2. Has the statistical analysis been performed appropriately and rigorously? 

Reviewer #1: No

Reviewer #2: Yes

3. Have the authors made all data underlying the findings in their manuscript fully available?

Reviewer #1: No

Reviewer #2: Yes

4. Is the manuscript presented in an intelligible fashion and written in standard English?

Reviewer #1: Yes

Reviewer #2: Yes

5. Review Comments to the Author

Reviewer #1: In this manuscript, authors evaluated the impacts of temperatures and precipitation on GDP across several industries in Chile by estimating region-industry panel data models. They found no statistically significant effect of precipitation changes on economic growth, but a negative impact of higher summer temperatures on ag-silviculture and fishing industries in Chile. This is an interesting study, but I have following comments on theoretical model, estimation methodology, result presentation and overall organization of the paper.

1. To me, the introduction section is not well organized and could be improved. I don't recall at the moment, but there must be more literature related to this topic than Colacito et al. 2019. Authors haven't specifically highlighted the rationale of the study and its contribution to the literature. Several paragraphs are related to results which usually should not be in the Intro section.

2. I haven't read Colacito et al. 2019 paper thoroughly, but what is the theoretical foundation (economic or other theory) of regressing GDP on temperature and ppt? Authors should explain how they did they come up with eq 3 as their econometric model. This is quite crucial.

3. Estimation methods: It appears that authors have set up the data in a panel framework (region & year), but they just employed OLS: how about fixed-effect, random-effect or other panel data estimation techniques? Why didn't you even try?

4) Result presentation: It would be way easier to follow the trend lines if authors presented Table 7-11 in graphs (line or area graphical presentation).

5)To me, conclusions and policy implications are also not strongly stated: what do the main results mean to the future of Chie and its economic growth? Based on your findings and projections, what are the insights/guidelines for policymakers and related industry leaders and stakeholder?

Reviewer #2: The authors in this manuscript investigate the impact of precipitation changes and temperature fluctuations over the period 1985-2017 in the Chilean case over 12 economic sectors. The paper is well written and despite several models offered to the reader, is able to provide straight conclusions. Nonetheless, I believe the manuscript could be further enhanced with the following suggestion listed below.

When the authors describe the data and where they retrieved them, they just mention the source (e.g., Central Bank of Chile, Chilean Bureau of Official Statistics, University of Delaware Air Temperature and Precipitation) without providing an effective reference in the references list. For completeness also these data sources should be properly mentioned in the references list. In this case also the effective date when data has been retrieved should be reported but I guess it could be not necessary.

The structure of your paper for sure is not “classical”, at least for my experience since an effective section for the literature review is missing while it is integrated within the Introduction section. Personally, this does not represent an issue as long as the Editor approves this structure.

There is a quite interesting study of the Standford University [1] which could be interested, and I guess it is worth mentioning, especially for either the Introduction or Conclusion section of your manuscript.

When you present the econometric model in section 2.3 you may add for completeness a classic reference for panel data models, such as the book of Baltagi [2].

You did not provide any preliminary analysis of your data, such as presence of autocorrelation, heterogeneity, cross-sectional dependency, or stationarity of your series. Since in your model you analyze GDP growth you should not have problems in terms of unit roots. Nonetheless, these issues are often undervalued in panel data analysis.

In your panel model you used robust standard errors clustered by region and year. However, are robust to which disturbance? I guess they are robust to heteroscedasticity and autocorrelation. Nonetheless, since you are working with regional data, maybe the sample could be affected also by cross-sectional dependency. This is an issue which could be addressed with time fixed effects or, for example, using proper robust standard errors able to take into account this disturbance. You may be interested in giving a look to Driskoll and Kraay robust standard errors, for example [3-4].

You performed your econometric model with both quarterly and monthly data. Since region-industry data is available only with yearly frequency, have you attempted to perform your model also with just yearly average data of temperature and precipitations?

Since you are dealing with regional data for Chile, maybe it could be an idea to enhance your manuscript also with a graphical regional representation of how each Chilean region – i.e., a choropleth map – contribute to the composition of the entire GDP of the country (considering a specific year of interest or an average of the time-range of your analysis) or for some specific sectors of interest (maybe you may add in the appendix the maps for all remaining sectors).

In the concluding section of your manuscript, you may consider some possible extension of your analysis, for example through the use of spatial [5-7] or dynamic time-series panel data models [8-9].

[1] Diffenbaugh, N. S., & Burke, M. (2019). Global warming has increased global economic inequality. Proceedings of the National Academy of Sciences, 116(20), 9808-9813.

[2] Baltagi, B. H. (2021). Econometric analysis of panel data. Springer Nature.

[3] Driscoll, J. C., & Kraay, A. C. (1998). Consistent covariance matrix estimation with spatially dependent panel data. Review of economics and statistics, 80(4), 549-560.

[4] Hoechle, D. (2007). Robust standard errors for panel regressions with cross-sectional dependence. The Stata Journal, 7(3), 281-312.

[5] Millo, G., & Piras, G. (2012). splm: Spatial panel data models in R. Journal of statistical software, 47, 1-38.

[6] Elhorst, J. P. (2014). Spatial panel data models. In Spatial econometrics (pp. 37-93). Springer.

[7] Belotti, F., Hughes, G., & Mortari, A. P. (2017). Spatial panel-data models using Stata. The Stata Journal, 17(1), 139-180.

[8] Chudik, A., Mohaddes, K., Pesaran, M. H., & Raissi, M. (2013). Debt, inflation and growth: robust estimation of long-run effects in dynamic panel data models. Cafe research paper, (13.23).

[9] Chudik, A., Mohaddes, K., Pesaran, M. H., & Raissi, M. (2018). Rising public debt to GDP can harm economic growth. Economic Letter, 13(3), 1-4.

6. PLOS authors have the option to publish the peer review history of their article (what does this mean?). If published, this will include your full peer review and any attached files.

Reviewer #1: No

Reviewer #2: No

---

## [Author Response · Author response to Decision Letter 0]

15 Feb 2022

Reply to the Editor and Journal requirements on "The impact of climate change on economic output across industries in Chile" PONE-D-21-33259

 Dear Editor Francisco Aguilar and Plos One journal office,

 Thank you for your report, suggestions and the two anonymous reviewers' feedback reports.

 Requirement 1: When submitting your revision, we need you to address these additional requirements.

 Reply: We now send the manuscript with the Plos One template and the correct author affiliations.

 Requirement 2: Please update your submission to use the PLOS LaTeX template. The template and more information on our requirements for LaTeX submissions can be found at http://journals.plos.org/plosone/s/latex.

 Reply: We now send the manuscript with the Plos One template.

 Requirement 3: We note that you have stated that you will provide repository information for your data at acceptance. Should your manuscript be accepted for publication, we will hold it until you provide the relevant accession numbers or DOIs necessary to access your data. If you wish to make changes to your Data Availability statement, please describe these changes in your cover letter and we will update your Data Availability statement to reflect the information you provide.

 Reply: We published the entire dataset in Stata (.dta) format online on the Mendeley repository:

 Madeira, C. (2022), "Panel data for the 15 Chilean regions with Weather and GDP variables", Mendeley Data, V1, doi: 10.17632/zyrdg56hzr.1.

 Requirement 4: Please amend your list of authors on the manuscript to ensure that each author is linked to an affiliation. Authors' affiliations should reflect the institution where the work was done (if authors moved subsequently, you can also list the new affiliation stating "current affiliation:…." as necessary).

 Reply: We now send the manuscript with the correct author affiliations.

 Additional Editor Comments:

 1) Two reviewers have offered constructive criticisms to the submitted manuscript, all of which shall be fully addressed. For instance, the authors should clearly state the contribution of this study and the rationale supporting their model specification. Within their econometric estimation, a justification for not using common approaches to panel data (e.g. fixed, random, mixed effects) is necessary. Preferably, and if relevant, such models should be run. The model seems to be later calibrated with US-based data, which should be properly motivated and explained within a Methods section.

 Reply: We justify the calibration with US-based data in the appendix, because the US is at the technological frontier and therefore may represent a better calibration for the future of Chile. Furthermore, the US has 50 states and longer panel data time series, therefore its calibration for the parameters can be less noisy than the shorter time-series and smaller regions of the Chilean dataset. This is justified in section 3.4 of the article. It is worth noting that the methodology applied for the US data by Colacito et al. (2019) is the same exact methodology we applied for Chile, therefore there is no methodological difference. Furthermore, the results with the US calibration are very similar and these are meant just as a robustness check to the main results.

 2) Editorial observations: Section 2 should be re-titled to 'Methods' or 'Methods and Data'. Other subtitles such as 'Results' should be simply labeled as such (the Results are for Chile as clearly stated from the Title, so there is no need to include 'Chile' in a subtitle). The language is adequate but some careful editing is needed. As a case in point, the authors write: "The analysis for the past 35 years would show that 85% of the economic activity..." In this case "past" should be avoided as there will likely be a mismatch between time of publication and the period covered in the study. The word "would" is not needed, as their econometrics results indeed show these trends. A revised text could read" Our analysis over the 1985-2017 period show that 85% of the economic activity...". Although this might seem trivial, it will help with readability and possible increase the impact of the manuscript.

 Reviewers' comments:

 Reviewer's Responses to Questions

 Reply: We followed your suggestion and checked again the manuscript for typos.

 Comments to the Author

 1. Is the manuscript technically sound, and do the data support the conclusions?

 Reviewer #1: Partly

 Reviewer #2: Yes

 Reply: We extended the manuscript with 6 new Tables and 5 new Figures plus additional analysis to support the conclusions.

 2. Has the statistical analysis been performed appropriately and rigorously?

 Reviewer #1: No

 Reviewer #2: Yes

 Reply: We added clarifying notes on the Introduction and section 2.3. We also added Tables 3, A1, A2, B4, B5, B6 and Figures 1, 2, 3, 4, 5, with additional statistical analysis.

 3. Have the authors made all data underlying the findings in their manuscript fully available?

 The PLOS Data policy requires authors to make all data underlying the findings described in their manuscript fully available without restriction, with rare exception (please refer to the Data Availability Statement in the manuscript PDF file). The data should be provided as part of the manuscript or its supporting information, or deposited to a public repository. For example, in addition to summary statistics, the data points behind means, medians and variance measures should be available. If there are restrictions on publicly sharing data---e.g. participant privacy or use of data from a third party---those must be specified.

 Reviewer #1: No

 Reviewer #2: Yes

 Reply: We published the entire dataset in Stata (.dta) format online on the Mendeley repository:

 Madeira, C. (2022), "Panel data for the 15 Chilean regions with Weather and GDP variables", Mendeley Data, V1, doi: 10.17632/zyrdg56hzr.1.

 4. Is the manuscript presented in an intelligible fashion and written in standard English?

 Reviewer #1: Yes

 Reviewer #2: Yes

 Reply: We reviewed again the manuscript for typos.

 5. Review Comments to the Author

 Reply: We provide two files with detailed replies to each reviewer.

 We formatted the article according to the requirements of the editor and the journal's office. We hope you are pleased and that our article is now ready to be accepted by the PLOS ONE. Kind regards,

 Karla Hernández

 Carlos Madeira

Reply to Reviewer 1 on "The impact of climate change on economic output across industries in Chile" PONE-D-21-33259

 Dear Colleague,

 Thank you for your report and suggestions. We are sending you a substantially revised draft of our manuscript, which includes adequate changes to account for all your comments and suggestions. For instance, you can easily check that the first draft had only 33 pages, while the current draft is 49 pages and represents therefore a more complete work that accounts for all the suggestions made by the journal. The new draft also has 6 entirely new tables and 5 entirely new Figures. We provide the paper in two versions. The first version is with the Plos One journal template and therefore it has just 21 pages. The second version which you find at the end of this reply to your reports is with the same template used in the first submission and therefore you can easily check it has 49 pages and substantially more material that answers your suggestions.

 In this letter we summarize how your suggestions were applied in changes to the manuscript. Our reply uses text in bold to emphasize the paragraphs or sections in which you can easily find the corresponding text modifications.

 Comment 1: To me, the introduction section is not well organized and could be improved. I don't recall at the moment, but there must be more literature related to this topic than Colacito et al. 2019. Authors haven't specifically highlighted the rationale of the study and its contribution to the literature. Several paragraphs are related to results which usually should not be in the Intro section.

 Reply: Most studies of climate change use data that is for many countries and only includes the national GDP. We use region-industry data for a single country and in this respect that is why we are more similar to Colacito et al. (2019), since other studies do not use data for specific industries. We clarify this by adding this text to the fourth paragraph of the Introduction: "Most studies for the impact of climate change on GDP use international level datasets with GDP for many countries and information on their temperatures and precipitation (Dell et al. 2012, Burke et al. 2015, Kahn et al. 2021). In this work, however, we use a dataset that is specific for Chile and its regions-industries. Therefore we apply a methodology similar to Colacito et al. (2019) who also use state-industry data specific to the USA, finding that higher summer temperatures affected negatively the economic output of at least half of the industries, especially finance, insurance and real estate.".

 In relation to the contribution of this article to the literature, we added this second paragraph in the Introduction: "This study provides a view of the economic impact of climate change in Chile over the past 35 years, focusing on its impact across different industries and regions. This presents a contribution relative to Colacito et al. (2019), who make a similar analysis for the USA across states and industries. Our work advances upon the previous literature by showing a similar analysis for Chile. Chile is an interesting case, because it is a developing economy with a much stronger relevance of the primary sectors in its output and it is located in the southern hemisphere which will be differently affected by climate change relative to the north (IPCC 2014, 2021)."

 In Economics articles it is usually standard to summarize the main results in the Introduction section and most Economics editors actually demand that the Introduction includes the main results. However, to fullfill your suggestion, we followed your comment and moved the sentences with the results to the Conclusions only. We think that since the Introduction was already quite long, then your suggestion is useful so that we wrote a shorter Introduction and that improves the readability of the article. The same information that we erased from the Introduction is written in even greater detail in the Conclusions, therefore your suggestion is appropriate and it improves the reading flow of the manuscript. In particular, we erased these paragraphs from the Introduction:

 1) "For instance, Agriculture is positively affected by the temperature increases during the month of November. (...) therefore the unavailability of regional-industry GDP data at a quarterly or monthly frequency makes statistical identification harder and casts some uncertainty on the interpretation of our findings."

 2) "Our estimates for the impact of the global climate change on the Chilean GDP growth rate in 2017 change between +0.1% and -0.2%, (...) especially because the most affected sectors (Agriculture and Fishing) represent just 4% of the national GDP."

 3) "Over time, the fraction of GDP represented by the sectors economically affected by climate change falls, (...) The stress test exercises are robust to using either the 2014 or the 2021 scenarios of the IPCC."

 Comment 2: I haven't read Colacito et al. 2019 paper thoroughly, but what is the theoretical foundation (economic or other theory) of regressing GDP on temperature and ppt? Authors should explain how they did they come up with eq 3 as their econometric model. This is quite crucial.

 Reply: We followed your suggestion and added this text to section 2.3 of the article:

 "It is well known that temperature affects the dynamics of virtually all chemical, biological and ecological processes (Burke et al. 2015), while precipitation can affect agriculture (Fernandes et al. 2012, Burke and Emerick (2016), especially in Latin America (Bárcena et al. 2019). and also non-agricultural activities if excessive floods disrupt transport and urban connections (Burke et al. 2015, Mendelsohn 2009). Chile, in particular, has been strongly affected in terms of reduced water availability (Gerten et al. 2011) and a decade long mega-drought (Hernández and Madeira 2021). Zivin and Neidell (2014) found that warmer temperatures reduce labor supply, while Cachon, Gallino, and Olivares (2012) document that high temperatures decrease productivity and performance.

 Seasonal temperatures and precipitation can affect productivity both in outdoor activities such as agriculture, fishing and construction (Mendelsohn 2009, Bárcena et al. 2019), but also for non-agricultural activities due to the influence of the weather on workers' health or urban movement (Burke et al. 2015, Colacito et al. 2019). For this reason our vector T_{r,s,t} for the measure of the weather variables in region r in season s of year t includes both average temperature and precipitation.

 There can be other shocks besides the weather (for instance, international shocks such as the Great Financial Crisis or higher demand from commodities due to a higher economic growth in China) that affect the economic growth of each industry i at time t. For this reason our chosen model must account for both time-industry fixed-effects (α_{t,i}) and the dynamic effect of shocks in the previous year by controlling for the lagged growth (Δy_{r,i,t-1}). Furthermore, an adequate model must account for regional heterogeneity in terms of natural resources, weather and industry specialization, therefore our model will include fixed-effects across regions and industries (α_{r,i}) and heterogeneous coefficients (β_{s,i} for the impact of the weather variables T_{r,s,t}, ρ_{i} for the impact of the lagged growth Δy_{r,i,t-1})."

 Comment 3: Estimation methods: It appears that authors have set up the data in a panel framework (region & year), but they just employed OLS: how about fixed-effect, random-effect or other panel data estimation techniques? Why didn't you even try?

 Reply: Actually, our model is a panel data model with fixed-effects, therefore this suggestion was already included in the original manuscript. OLS means that the model is linear, but it can include fixed-effects. Therefore OLS is a class of linear models that includes Panel Data models with fixed-effects. Most panel data models with fixed-effects are estimated by OLS. The random-effects require Maximum Likelihood Estimation (MLE). To help clarify this point we added this text to the top of Table 4 and Table 5 with the econometric model estimates "OLS with fixed-effects by time and region, separate regressions by industry". Therefore it is clear that there are different coefficients for each industry (this includes different Betas plus different variances) and also fixed-effects by time-industry and region-industry.

 We also added this text to the third paragraph of the Introduction, which explains why we estimated panel data with fixed-effects rather than random-effects: "Our econometric model has different coefficients for each industry and it includes as control variables the temperature and precipitation for each season (whether quarterly seasons or months) plus the industry-region growth lag, time fixed-effects at the year level, and fixed-effects for the regions. The model therefore accounts for both unobserved macroeconomic shocks affecting each industry and unobserved heterogeneity at the region-industry level."

 We also added this text to the last paragraph of section 2.3: "In relation to other alternatives such as random-effects, the fixed-effects added in our model help to control for fixed unobservables across time-industry and region-industry without imposing any distribution assumption or any correlation assumption with the other observable variables, while the random-effects models assume that the fixed unobservable errors are normal distributed and uncorrelated with the other observable variables (Baltagi 2021). It is also worth noting that several of the previous papers that estimate the impact of climate change on GDP use fixed-effects rather than random-effects (see Dell et al. 2012, Burke et al. 2015, Colacito et al. 2019, Kahn et al. 2021)."

 Comment 4: Result presentation: It would be way easier to follow the trend lines if authors presented Table 7-11 in graphs (line or area graphical presentation).

 Reply: We think that the line graphs would be too confusing and difficult to read. For one, there are 12 industries and therefore there would be too many lines intersecting each other. Also, the scale since some industries would be decreasing almost to -100% and other industries would be increasing by values even larger than 100%. We think therefore the Tables are easier to read than Figures and the readers can look at the numbers and see the exact values rather than trying to guess the values on a big graphical scale that changes between -100% and values larger than +100%.

 Comment 5: To me, conclusions and policy implications are also not strongly stated: what do the main results mean to the future of Chie and its economic growth? Based on your findings and projections, what are the insights/guidelines for policymakers and related industry leaders and stakeholders?

 Reply: Thank you for the suggestion. We added this as the final paragraph of the Conclusions:

 "One policy implications of this work is that more research is required for knowing whether the effects of climate change are permanent over the long term or not. Our work finds an impact of seasonal temperatures on the growth rate of the GDP of several industries and an effect on the growth rate may have large accumulated impacts over several years, as shown in our exercises for Chile and previous studies for the USA (Colacito et al. 2019). However, many industries may undertake investments to mitigate the effects of climate change, such as finding alternative energy sources or crops that are better suited to warmer weather (Olmstead and Rhode 2011). Governments can also implement new regulations and build better infrastructure to adjust for the long run climate. Recent research, however, has found evidence of fairly negative effects of climate change on agriculture even after several decades (Hornbeck 2012, Burke and Emerick 2016), showing that current adjustments may not be enough to mitigate the negative shock of global warming. It is therefore crucial for economic research to provide greater evidence on all the possible short-run and long-run effects of climate change on different industries and natural resources (Albagli 2021) in order to evaluate the value of environmental regulations and green investments (Hoffmann et al. 2020)."

 We also formatted the article according to the requirements of the editor, the journal's office and the other anonymous reviewer. We hope you are pleased and that our article is now ready to be accepted by the PLOS ONE. Kind regards,

 Karla Hernández

 Carlos Madeira

Reply to Reviewer 2 on "The impact of climate change on economic output across industries in Chile" PONE-D-21-33259

 Dear Colleague,

 Thank you for your report and suggestions. We are sending you a substantially revised draft of our manuscript, which includes adequate changes to account for all your comments and suggestions. For instance, you can easily check that the first draft had only 33 pages, while the current draft is 49 pages and represents therefore a more complete work that accounts for all the suggestions made by the journal. The new draft also has 6 entirely new tables and 5 entirely new Figures. We provide the paper in two versions. The first version is with the Plos One journal template and therefore it has just 21 pages. The second version which you find at the end of this reply to your reports is with the same template used in the first submission and therefore you can easily check it has 49 pages and substantially more material that answers your suggestions.

 In this letter we summarize how your suggestions were applied in changes to the manuscript. Our reply uses text in bold to emphasize the paragraphs or sections in which you can easily find the corresponding text modifications.

 Comment 1: When the authors describe the data and where they retrieved them, they just mention the source (e.g., Central Bank of Chile, Chilean Bureau of Official Statistics, University of Delaware Air Temperature and Precipitation) without providing an effective reference in the references list. For completeness also these data sources should be properly mentioned in the references list. In this case also the effective date when data has been retrieved should be reported but I guess it could be not necessary.

 Reply: We followed your suggestion and added the references to to the Central Bank of Chile, Chilean Bureau of Official Statistics, University of Delaware Air Temperature and Precipitation to the Reference list. We also published the complete dataset we used on Mendeley and added it to the References list.

 References added:

 Central Bank of Chile (2019), "Base de Datos Estadísticos: Cuentas Nacionales por región, Series de tiempo de precios de UF," accessed on September of 2019, Banco Central de Chile.

 Madeira, C. (2022), "Panel data for the 15 Chilean regions with Weather and GDP variables", Mendeley Data, V1, doi: 10.17632/zyrdg56hzr.1.

 University of Delaware (2019), "University of Delaware Air Temperature and Precipitation," accessed on October of 2019.

 Comment 2: The structure of your paper for sure is not "classical", at least for my experience since an effective section for the literature review is missing while it is integrated within the Introduction section. Personally, this does not represent an issue as long as the Editor approves this structure.

 Reply: We kept the structure of the article with the Literature Review as part of the Introduction. Like you, we have no personal taste on this. It is just a matter of divergent views in the academic world that have not yet settled their views about the role of the Literature Review, but many editors prefer that the literature review should be limited to just 1 or 2 paragraphs in the Introduction.

 To account for your comment, we also added this sentence to the Introduction which makes reference to a previous work of ours which presents a very exhaustive literature review for Chile on the penultimate paragraph of the Introduction: "Finally, Hernández and Madeira (2021) show a literature review about the impact of climate change in Chile in a wide range of aspects, from GDP to water availability and migration."

 Comment 3: There is a quite interesting study of the Stanford University [1] which could be interested, and I guess it is worth mentioning, especially for either the Introduction or Conclusion section of your manuscript.

 Reply: Thank you for mentioning this study of Diffenbaugh and Burke 2019. We followed your comment and added two sentences on the first paragraph of the Introduction:

 "Due to its worst impact on the poorest countries (Diffenbaugh and Burke 2019) and the poorest households, climate change will be a significant threat to economic growth and reducing income inequality in Latin American countries (Bárcena et al. 2019, Cavallo and Hoffmann 2020). Empirical estimates show that global warming reduced the GDP per capita of the poorest countries by 17-31% over the last half century, making it more difficult for poorer nations to converge towards developed economies and increasing inequality between countries (Diffenbaugh and Burke 2019)."

 Comment 4: When you present the econometric model in section 2.3 you may add for completeness a classic reference for panel data models, such as the book of Baltagi [2].

 Reply: We followed your suggestion and added the reference to Baltagi [2] on the last paragraph of section 2.3 and also added the reference of Wooldridge (2010):

 "In relation to other alternatives such as random-effects, the fixed-effects added in our model help to control for fixed unobservables across time-industry and region-industry without imposing any distribution assumption or any correlation assumption with the other observable variables, while the random-effects models assume that the fixed unobservable errors are normal distributed and uncorrelated with the other observable variables (Baltagi 2021, Wooldridge 2010). It is also worth noting that several of the previous papers that estimate the impact of climate change on GDP use fixed-effects rather than random-effects (see Dell et al. 2012, Burke et al. 2015, Colacito et al. 2019, Kahn et al. 2021)."

 Furthermore, section 5.1 of the appendix again mentions Baltagi (2021) and Wooldridge (2010) to explain our panel data tests for heterocedasticity and unit roots.

 Comment 5: You did not provide any preliminary analysis of your data, such as presence of autocorrelation, heterogeneity, cross-sectional dependency, or stationarity of your series. Since in your model you analyze GDP growth you should not have problems in terms of unit roots. Nonetheless, these issues are often undervalued in panel data analysis.

 Reply: We followed your suggestion and included an entirely new section 5.1 in the appendix to implement panel data tests for heterocedasticity and unit roots of the real growth rate of each industry (Δy_{r,i,t}), according to the methodologies suggested in Baltagi (2021) and Wooldridge (2010). Table A.1 rejects the hypothesis of homocedasticity, which justifies our option for standard-errors clustered by region and year in section 3.1. Table A.2 rejects the null hypothesis of unit roots in the panel data, which justifies the option in our model in section 2.3 for not considering a unit root.

 Comment 6: In your panel model you used robust standard errors clustered by region and year. However, are robust to which disturbance? I guess they are robust to heteroscedasticity and autocorrelation. Nonetheless, since you are working with regional data, maybe the sample could be affected also by cross-sectional dependency. This is an issue which could be addressed with time fixed effects or, for example, using proper robust standard errors able to take into account this disturbance. You may be interested in giving a look to Driskoll and Kraay robust standard errors, for example [3-4].

 Reply: We followed your suggestion and added the regressions with the Driskoll and Kraay robust standard errors to Table B4 and Table B5 in section 5.2 of the appendix.

 Comment 7: You performed your econometric model with both quarterly and monthly data. Since region-industry data is available only with yearly frequency, have you attempted to perform your model also with just yearly average data of temperature and precipitations?

 Reply: We followed your suggestion. We added Table B6 in section 5.2 of the appendix with this analysis of yearly weather. Most yearly weather variables are not statistically significant and do not show the same coefficients, which is expected due to yearly weather hiding shocks to seasonal temperature (Colacito et al. 2019).

 Comment 8: Since you are dealing with regional data for Chile, maybe it could be an idea to enhance your manuscript also with a graphical regional representation of how each Chilean region -- i.e., a choropleth map -- contribute to the composition of the entire GDP of the country (considering a specific year of interest or an average of the time-range of your analysis) or for some specific sectors of interest (maybe you may add in the appendix the maps for all remaining sectors).

 Reply: We followed your suggestion. We added section 5.5 in the appendix with 2 new figures for the regional GDP across the entire country. Figure 4 shows the share of the GDP (in %) across regions in Chile, according to the averages between 1985 and 2017. Figure 5 shows the same values of regional GDP in the most recent year of 2017, confirming that there was no difference in terms of the economic importance of each region in recent years.

 Note also that in section 2.1 we added Table 2 with the fraction of each industry for the GDP of each region. We also added the Figure 1 in section 2.2 with the minimum, mean, maximum of the variables for temperature and precipitation with different weights for each region. Figure 2 in section 2.2 reports the same variables across 4 macro-regions (North, Central, South, Metropolitan Capital). Table 3 in section 2.2 then reports the temperature and precipitation changes between 1950 and 2017 for the same macro-regions (weighted by surface area). Section 5.4 in the appendix then shows Figure 3 which reports the temperature and precipitation changes between 1950 and 2017 for the same macro-regions (weighted by regional GDP).

 Comment 9: In the concluding section of your manuscript, you may consider some possible extension of your analysis, for example through the use of spatial [5-7] or dynamic time-series panel data models [8-9].

 Reply: We followed part of your suggestion. We are not so experienced with spatial models, therefore we thought it was a risky option to improvise and learn such methods in just a few days to revise the manuscript. However, we point out that the model we propose in equation 3) of section 2.3 is already a dynamic panel data model, which has fixed-effects by region and time and heterogeneous coefficients by industry. We expanded section 2.3 to include both more comments about the economic intuition of the variables of the model and also a better justification for the panel-data econometric methodology we applied:

 "Our econometric model has different coefficients for each industry and it includes as control variables the temperature and precipitation for each season (whether quarterly seasons or months) plus the industry-region growth lag, time fixed-effects at the year level, and fixed-effects for the regions. The model therefore accounts for both unobserved macroeconomic shocks affecting each industry and unobserved heterogeneity at the region-industry level."

 "In relation to other alternatives such as random-effects, the fixed-effects added in our model help to control for fixed unobservables across time-industry and region-industry without imposing any distribution assumption or any correlation assumption with the other observable variables, while the random-effects models assume that the fixed unobservable errors are normal distributed and uncorrelated with the other observable variables (Baltagi 2021). It is also worth noting that several of the previous papers that estimate the impact of climate change on GDP use fixed-effects rather than random-effects (see Dell et al. 2012, Burke et al. 2015, Colacito et al. 2019, Kahn et al. 2021)."

 We also formatted the article according to the requirements of the editor and the journal's office. We hope you are pleased and that our article is now ready to be accepted by the PLOS ONE. Kind regards,

 Karla Hernández

 Carlos Madeira

---

## [Decision Letter · Decision Letter 1]

29 Mar 2022

The impact of climate change on economic output across industries in Chile

PONE-D-21-33259R1

Dear Dr. Madeira,

We’re pleased to inform you that your manuscript has been judged scientifically suitable for publication and will be formally accepted for publication once it meets all outstanding technical requirements.

Kind regards,

Carla Pegoraro

Division Editor

PLOS ONE

Reviewers' comments:

Reviewer's Responses to Questions

**Comments to the Author**

1. If the authors have adequately addressed your comments raised in a previous round of review and you feel that this manuscript is now acceptable for publication, you may indicate that here to bypass the “Comments to the Author” section, enter your conflict of interest statement in the “Confidential to Editor” section, and submit your "Accept" recommendation.

Reviewer #1: All comments have been addressed

Reviewer #2: All comments have been addressed

2. Is the manuscript technically sound, and do the data support the conclusions?

Reviewer #1: Yes

Reviewer #2: Yes

3. Has the statistical analysis been performed appropriately and rigorously? 

Reviewer #1: Yes

Reviewer #2: Yes

4. Have the authors made all data underlying the findings in their manuscript fully available?

Reviewer #1: No

Reviewer #2: Yes

5. Is the manuscript presented in an intelligible fashion and written in standard English?

Reviewer #1: Yes

Reviewer #2: Yes

6. Review Comments to the Author

Reviewer #1: While authors addressed all of my comments in this version, I still think pooled OLS is different from fixed and random effect models. I saw authors already cited Wooldrige's book; please see his explanation of panel data models and similarly Greene's book also has it in detail.

Similarly, I still think figures presenting only meaningful results with various color combinations are way better in results presentation, compared to tables with myriad numbers. After all, we are looking at the projections and trends; exact % numbers are less relevant.

Having said this, I look forward to seeing this paper published in PLOS One.

Reviewer #2: The author(s) addressed all my comments. However, I would like to add two small possible enhancements:

1) You may refer in your manuscript to your Appendix analysis, such as the various test you performed or the model(s) performed with different standard errors.

2) In the Appendix, when you show the results of your model with Driscoll and Kraay robust standard errors you should not just show the table(s) but also provide some small comments of how results may differ from those showed in the main body of your manuscript.

7. PLOS authors have the option to publish the peer review history of their article (what does this mean?). If published, this will include your full peer review and any attached files.

Reviewer #1: No

Reviewer #2: No

---

## [Editor Report · Acceptance letter]

7 Apr 2022

PONE-D-21-33259R1 

The Impact of Climate Change on Economic Output across Industries in Chile 

Dear Dr. Madeira:

I'm pleased to inform you that your manuscript has been deemed suitable for publication in PLOS ONE. Congratulations! Your manuscript is now with our production department. 

Kind regards, 

on behalf of

Dr Carla Pegoraro 

Staff Editor

PLOS ONE